# Small soluble α-synuclein aggregates are the toxic species in Parkinson's disease

Derya Emin [1,2], Yu P. Zhang[1,2], Evgeniia Lobanova[1,2], Alyssa Miller[1], Xuecong Li [3,4], Zengjie Xia [1,2], Helen Dakin [1,2], Dimitrios I. Sideris[1], Jeff Y. L. Lam [1,2], Rohan T. Ranasinghe[1], Antonina Kouli[5], Yanyan Zhao[6], Suman De [1,7], Tuomas P. J. Knowles [1], Michele Vendruscolo [1], Francesco S. Ruggeri [1,3,4], Franklin I. Aigbirhio[6], Caroline H. Williams-Gray [5] & David Klenerman [1,2] ✉

Soluble α-synuclein aggregates varying in size, structure, and morphology have been closely linked to neuronal death in Parkinson's disease. However, the heterogeneity of different co-existing aggregate species makes it hard to isolate and study their individual toxic properties. Here, we show a reliable non-perturbative method to separate a heterogeneous mixture of protein aggregates by size. We find that aggregates of wild-type α-synuclein smaller than 200 nm in length, formed during an in vitro aggregation reaction, cause inflammation and permeabilization of single-liposome membranes and that larger aggregates are less toxic. Studying soluble aggregates extracted from post-mortem human brains also reveals that these aggregates are similar in size and structure to the smaller aggregates formed in aggregation reactions in the test tube. Furthermore, we find that the soluble aggregates present in Parkinson's disease brains are smaller, largely less than 100 nm, and more inflammatory compared to the larger aggregates present in control brains. This study suggests that the small non-fibrillar α-synuclein aggregates are the critical species driving neuroinflammation and disease progression.

Parkinson's disease (PD) is the second most common neurodegenerative disorder after Alzheimer's disease, affecting up to 3% of the population over the age of 65[1]. PD is initially characterized by the irreversible damage to dopaminergic neurons in the substantia nigra that correlates with motor symptoms including bradykinesia, rigidity, and tremor; but more widespread pathology in limbic and cortical regions evolves over the disease course and is linked to a range of non-motor features[2–4]. Post-mortem brain analysis has shown the presence of proteinaceous inclusions, often referred to as Lewy bodies, which primarily contain aggregated α-synuclein[5]. Although, a recent paper

identified disintegrated membranes and organelles as the main constituents of Lewy bodies rather than α-synuclein[6]. Lewy bodies are found within the substantia nigra as well as in other subcortical and cortical regions to a variable extent[7]. Their presence in limbic and cortical regions correlates with the development of PD-associated dementia[8]. During an aggregation reaction, α-synuclein monomers convert intracellularly into various sizes, structures, and morphologies over time[9–12]. All these different α-synuclein species coexist, making it challenging to identify the toxic species in complex human biofluids and tissues[13]. Previous studies have shown the small soluble prefibrillar

[1]Yusuf Hamied Department of Chemistry, University of Cambridge, Lensfield Road, Cambridge, UK. [2]UK Dementia Research Institute, University of Cambridge, Cambridge, UK. [3]Laboratory of Organic Chemistry, Wageningen University and Research, Wageningen, Netherlands. [4]Physical Chemistry and Soft Matter, Wageningen University and Research, Wageningen, Netherlands. [5]John van Geest Centre for Brain Repair, Department of Clinical Neurosciences, University of Cambridge, Cambridge, UK. [6]Molecular Imaging Chemistry Laboratory, Wolfson Brain Imaging Centre, Department of Clinical Neurosciences, University of Cambridge, Cambridge, UK. [7]Sheffield Institute for Translational Neuroscience, University of Sheffield, Sheffield, UK. ✉e-mail: dk10012@cam.ac.uk

aggregates of α-synuclein, often referred to as oligomers, rather than mature fibrils, are the most cytotoxic aggregate species in vitro[11,14–16]. There is no rigorous definition of oligomers but can include any soluble aggregate of an intermediate size between monomers and insoluble fibrils[17].

Oligomers only account for a small fraction of the total intercellular α-synuclein but are of great interest as they are known to be involved in several different pathways leading to cell death and neurodegeneration. Some of these mechanisms include disruption of the cell membrane[14,18,19], synaptic loss[20–23], dysfunctions of the endoplasmic reticulum and mitochondria[24,25], seeding capacity[14,26–28], and neuroinflammation[22,29–31]. However, other work has shown that while oligomers cause cell death to dopaminergic neurons, fibrils, which are capable of seeding aggregates, are more toxic and lead to sustained progressive loss of neurons[32,33]. Many of these experiments are performed using acute doses of aggregates so it is unclear which mechanisms of toxicity will be most important at physiological concentrations of aggregates where the distribution of aggregate sizes might differ. Therefore, despite various efforts, key questions regarding how toxicity changes with size and which distinct species form in humans remain unanswered. So far, the generation and stabilization of toxic in vitro oligomers has been challenging. Previously published work has either made use of trapped oligomers[16,34], the addition of stabilizers or cross-linkers[14,35–39], high concentrations or mutated versions of α-synuclein[11,32]. Despite these major advances, a crucial unanswered question is whether these lab-generated oligomers have different properties to the actual aggregates found in the brain. In this work, we have used a centrifugation-based method to isolate different species of α-synuclein. A whole range of differently structured aggregates are extracted and can be compared to each other. This method is minimally perturbative and does not need the addition of external reagents or the use of mutants, enabling the study of toxicity as a function of size. Previous publications on tau and Aβ sucrose gradient fractionation have given some insight into the structure of the potentially toxic oligomers[40,41].

While many previous studies characterizing α-synuclein aggregates by size have been conducted in vitro, more research remains to be done on identifying the soluble aggregates present in the human brain. Endogenous levels of α-synuclein aggregates were measured in the substantia nigra in control and PD post-mortem tissue by Je et al. at the single-molecule level[42]. They reported that the oligomeric species accounted for 56% of the α-synuclein species in the PD samples but only 23% in non-PD control samples. The next step is the study of the toxicity of soluble aggregates from PD brain tissue.

In this study, we focus on identifying the α-synuclein aggregate-induced toxicity depending on aggregate size and morphology. Therefore, we use a density centrifugation method to separate an aggregation mixture by size and characterize the structure of differently sized aggregates using high-resolution imaging techniques. Later, we follow their toxic properties using two well-established assays: one based on the ability of the aggregates to disrupt cell membranes, and the second based on the aggregate-induced inflammatory response of BV2 microglial-like cells. We also compare these fractionated aggregates to extracted soluble aggregates from human post-mortem PD and control brains, in terms of size and toxicity, to determine the aggregate characteristics that are most closely linked to the disease state.

## Results

### Structural characterization of α-synuclein fractions

Producing a distinct aggregate species with specific size and morphology has been challenging due to the heterogeneity inherent in the aggregation process. To generate a more homogeneous aggregate population, we utilized a sucrose gradient approach, which is based on the principle that proteins can be separated by mass by accumulating in different sucrose concentrations during centrifugation. Therefore, we mixed α-synuclein from different time points taken from an aggregation reaction at 37 ˚C during constant shaking and loaded this mixture onto a discontinuous gradient with 10, 20, 30, 40 and 50% sucrose concentrations (Fig. 1A). Using transmission electron microscopy (TEM), we identified that most of the small structured α-synuclein aggregates could be found in 20% fraction, with predominantly spherical species present, which started to get more elongated in the 30% fraction. The 40 and 50% fractions consisted mainly of fibrillar species with varying lengths. The abundance of aggregates in the 10% fractions was very low, pointing out that this fraction might largely consist of α-synuclein monomer. (Fig. 1B).

α-synuclein aggregates could be detected in the 20–50% sucrose fractions using total internal reflection microscopy (TIRFM) with the amyloid-specific fluorescent dye thioflavin T (ThT) (Fig. 2). Aggregates in the 10% fraction could not be detected using ThT, pointing to the fact that there is minimal β-sheet containing species in this fraction (Supplementary Fig. 1). This method allowed us to determine and compare the fluorescence intensity of each aggregate (Fig. 2 and Supplementary Table 1). The data show a gradual increase in the median intensity from low to high sucrose concentrations (Fig. 2A–D) accompanied by an apparent shift in size as the small elliptical species predominantly present in the 20 and 30% fractions become larger and turn into elongated fibrils detectable in the 40% and 50% fraction. Using the Kolmogorov-Smirnov test, the p-values show that the corresponding intensity from each fraction is significantly different from the previous one confirming an increase in size.

### Length distribution of α-synuclein sucrose fractions

We used the fluorescent dye thioflavin X (ThX), a derivative of the widely used dye thioflavin T with enhanced photophysical properties, to study small structural details through transient amyloid binding[43]. By taking advantage of its transient nature, we could perform single-molecule super-resolution microscopy of α-synuclein aggregates with a precision of 20 nm[44]. Similarly to ThT, the 10% fraction could not be super-resolved using ThX due the minimal presence of β-sheet aggregates. We observe a clear trend indicating a gradual increase in length (Fig. 3A, B). The 20% fraction has an average length of $190 \pm 30$ nm, the 30% fraction of $240 \pm 30$ nm, the 40% of $290 \pm 30$ nm and the 50% of $390 \pm 90$ nm. The differences in the length distribution are statistically significant. The super-resolution representative images (Fig. 3D–G) also corroborate a clear size change.

Morphology is another parameter that can be analysed to understand aggregate structure. Using our super-resolution imaging, we investigated the eccentricity of the aggregates in the different fractions (Fig. 3C). The eccentricity of a cluster was determined by fitting an ellipse to the cluster and determining the focal distance of the ellipse divided by the maximum distance of the major axis. Values close to 1 indicate a fibrillar morphology, whereas a perfect circle has a value of 0. The values of the eccentricity of aggregates is typically between 0.3 and 1. We observed a change in eccentricity in the different sucrose fractions, although it was not statistically significant, the aggregates became more fibrillar in structure with an increasing percentage of sucrose. The 40% and 50% fractions had higher eccentricities, reaching 0.93 in the 50% fraction, while the 20% and 30% fractions had eccentricities around 0.84.

### Analysis of structural changes using AFM and antibodies

In order to analyze and unravel the 3D structure of the aggregates present in each sucrose fraction, we used high-resolution and phase-controlled AFM imaging, which is able to resolve structures with angstroms resolution including monomers (Fig. 4A–E)[45–47]. Samples for AFM analysis were deposited on a negative mica surface which is ideal to study soluble aggregates and oligomers. However, mica has less affinity to larger aggregates although some larger aggregates are

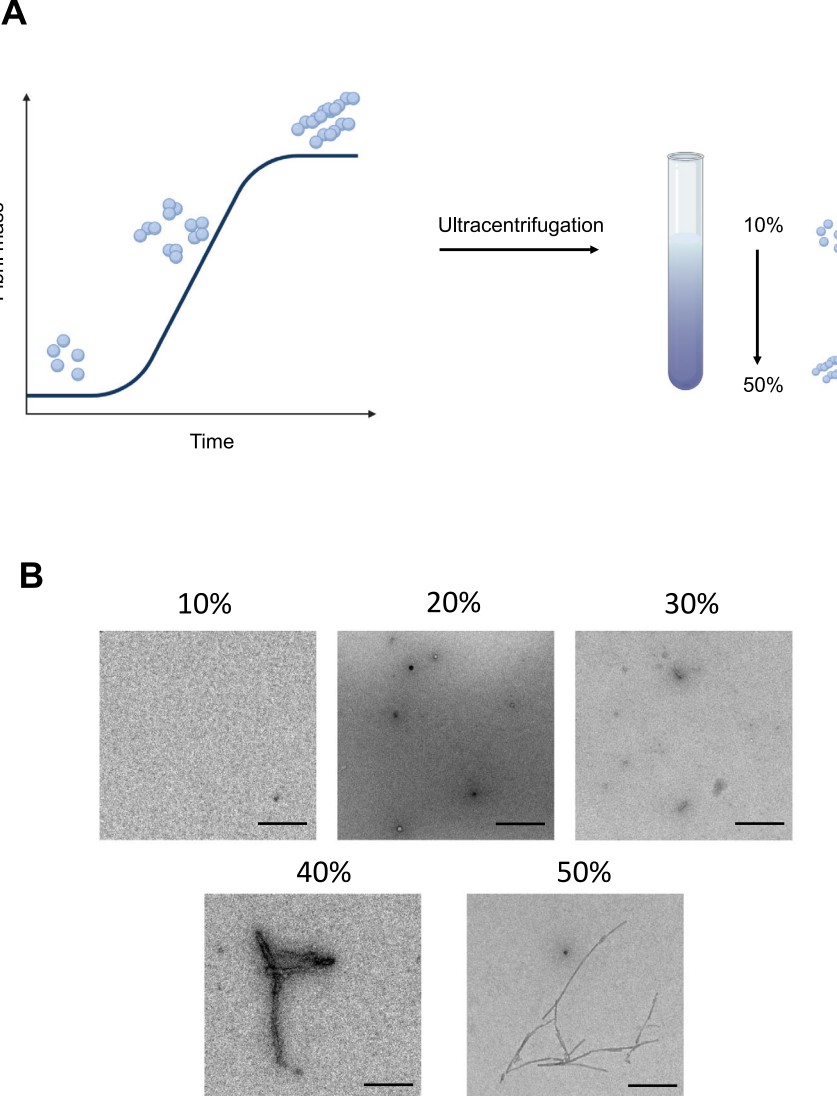

**Fig. 1 | Principle of protein separation by sucrose density centrifugation.**
**A** Schematic representation of sucrose density centrifugation for a protein mixture. In vitro α-synuclein is aggregated and at specific time points aliquots are taken from the aggregation reaction. All time points are combined in a sucrose solution with increasing sucrose concentration (10, 20, 30, 40, and 50%, from top to bottom).

The schematic has been created with BioRender.com. **B** Representative transmission electron micrograph images of aggregates present in individual fractions after density centrifugation. Scale bar = 200 nm. One out of two independent replicates represented here. Images were representative across experiments.

visible (Supplementary Fig. 2)[47–49]. We found a similar trend to a previously published report characterizing Aβ with sucrose gradient[40]. There is a net increase in the height and the cross-sectional diameter of the aggregates present in each fraction, with the 50% fraction containing the largest aggregates (Fig. 4F, G). Compared to the higher density fractions, the 10 and 20% fractions have predominantly monomeric protein (up to 0.5 nm in height)[47]. The 20 and 30% both showed the presence of protofilaments with a height of 0.6–0.8 nm. In the 40 and 50% fractions, larger mature amyloid aggregates with a height of up to 6 nm were identified (Supplementary Fig. 2). While these data were in qualitative agreement with the super-resolution imaging of aggregates, it also showed that AFM imaging was more sensitive towards smaller aggregates and monomers present in the fractions that were less accessible to ThX super-resolution imaging.

Antibody-based imaging is a powerful tool to study the surface accessibility of specific aggregates. Given that the super-resolution imaging showed an increase in length from the 20 to 50% sucrose fractions to gain further structural insight, we characterized aggregate structure with two different α-synuclein antibodies (Fig. 5). We used,

MJF, a conformation-specific antibody recognizing aggregated/filamentous species, and SC, a sequence-specific antibody targeting amino acids 121–125, using methods adapted from a previously published study[42]. The number of α-synuclein aggregates detected with the SC antibody, which targets a C-terminus epitope, was greater in the lower sucrose fractions and smallest in the 40-50% fractions, indicating that the C-terminus is more accessible in the smaller aggregates and enclosed in fibrils, which is in agreement with previously published NMR data[50]. The MJF antibody did not detect any aggregated α-synuclein in the 10% fraction, confirming a lack of β-sheet-rich structures in this fraction. A modest number of aggregates was instead detected using this antibody in the 20–40% fractions, with maximum quantity in the 50% fraction, which consists mainly of fibrils based on the SR data. These results are in agreement with our previous data (Figs. 2 and 3). To confirm that the number of detected species is due to the specific binding of the detection antibody to the aggregate, we compared the binding of an isotype antibody control (IgG) to the fractions (Supplementary Fig. 3). The number of detected spots decreased for the case of the labelled isotype control by a factor of ~20

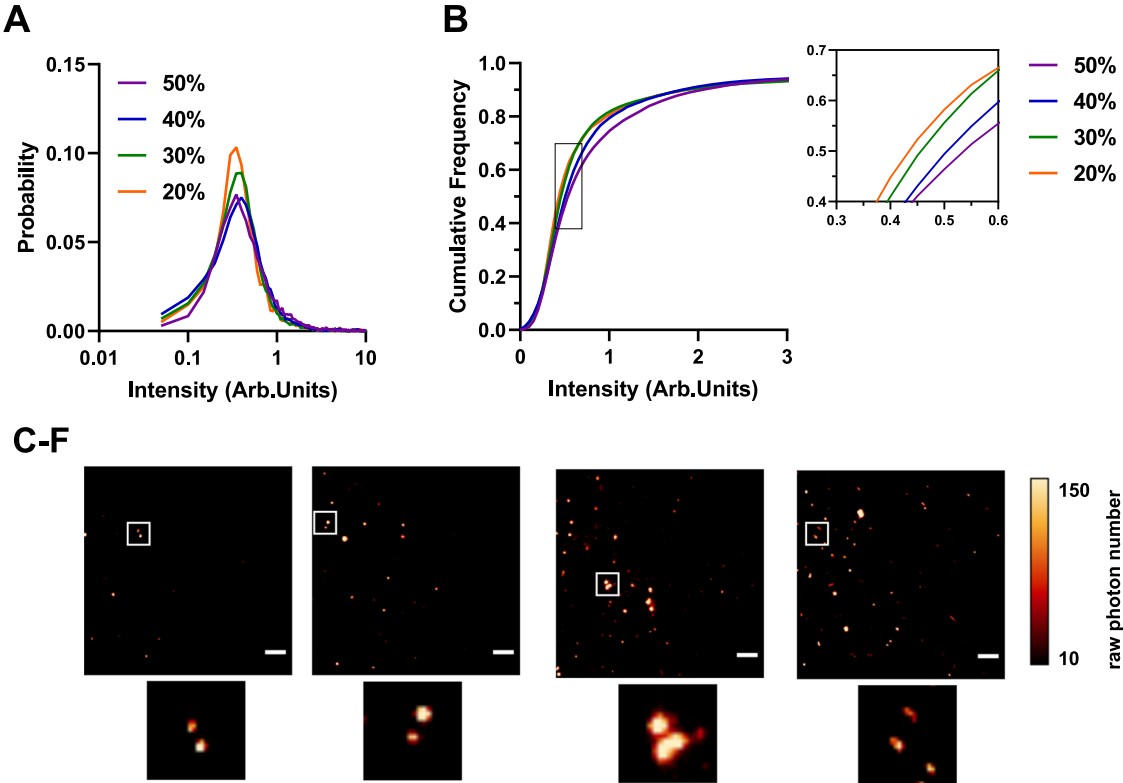

**Fig. 2 | Single-molecule characterization of differently sized α-synuclein aggregates using thioflavin T. A** Histogram of the thioflavin T (ThT) fluorescence intensity distributions for different individual fractions Each histogram is the collected data from three independent gradient centrifugations. **B** Cumulative frequency distribution for ThT intensity for 20–50% sucrose fractions. Using the Kolmogorov-Smirnov test, the *p*-values show that the corresponding intensity from each fraction is significantly different from the previous one confirming an increase in size. Kolmogorov-Smirnov test with 20% vs 30% $p < 0.001$, 30% vs 40% $p < 0.001$

and 40% vs 50% $p < 0.001$. (20% $N = 3378$, 30% $N = 8296$, 40% $N = 11,812$, 50% $N = 11,816$). **C–F** Representative TIRF images for each fraction including individual zoom-ins. The images were cropped and the contrast was adjusted for clarity (Scale bar = 5 μm). The LUT NanoJ-Orange has been applied to images with a calibration bar. One out of three representative replicates is displayed here. Images were representative across experiments. Each individual colour is representing a distinct sucrose fraction (orange = 20%, green = 30%, blue = 40%, violet = 50%). Source data are provided as a Source Data file. Arb.Units arbitrary units.

for SC and about 30 for MJF, confirming that the signal is due to specific antibody-antigen interactions.

### The cytotoxic properties of aggregates correlates with their size

To understand how the sizes of the individual aggregates influence their cytotoxic properties, each sucrose fraction was diluted to a total monomer concentration of 500 nM and added to BV2 mouse microglia cells for 24 h. Previous work shows that α-synuclein aggregates are stable once formed, even upon dilution to picomolar concentrations[9], and we have previously found no change in the number of aggregates under these conditions[29,51]. Afterwards, the release of tumour necrosis factor α (TNFα), a pro-inflammatory cytokine, strongly implicated in the progression of PD[52], was quantified after 24 h upon incubation with the fractions (Fig. 6A, C). The TNFα response was greatest upon stimulation with the 20% fraction. The 20% fraction accounts for sub-diffraction small species with an average length of 190 nm as determined by super-resolution microscopy (Fig. 3A). The larger fibrillar species present in the 40 and 50% fractions caused the lowest neuroinflammatory response, while the species in the 30% fraction produced an intermediate response. The aggregates in the 10% fraction are smaller than the species in the 20% according to the AFM analysis (Fig. 4F, G) but produced a similar inflammatory response to the 20% fraction. Hence, overall the data suggest that smaller non-fibrillar aggregates are more inflammatory than larger fibrillar aggregates. The fractions were then added to liposomes and their ability to disrupt lipid membranes and cause calcium influx was assessed (Fig. 6B, D). In this experiment, the aggregates present in the 10 and 20% fractions

caused more membrane disruption than the larger species present in the 30–50% fractions. Overall, these results support the hypothesis that small soluble aggregates rather than fibrils cause greater toxicity.

### Smaller aggregates of α-synuclein are found in PD brains and are more inflammatory

Soluble aggregates were extracted from post-mortem human brain tissue according to a previously published protocol (Supplementary Fig. 4)[53]. Briefly, brain pieces were cut to 300 mg and incubated in the buffer for 1.5 h. Afterwards, the upper 90% was centrifuged. Following centrifugation, the remaining upper 90% was centrifuged again. In the last step, the extracted aggregates were dialysed for 72 h in order to remove any pro-inflammatory cytokines. We focused on the amygdala as the region of interest as recent clinicopathological work has demonstrated that both α-synuclein pathology and neuroinflammation in this region are correlated and associated with the development of PD dementia[8]. We compared soluble protein extracts from the amygdala from three PD cases and three similarly aged controls without neurological disease during life (Table 1). Immunohistochemically analysis of formalin-fixed sections confirmed Lewy body pathology in the amygdala in all three PD cases and none of the controls (Supplementary Fig. 5). The aggregate length was determined by aptamer paint (AD-PAINT) (Fig. 7), a super-resolution method that uses an aptamer for aggregate recognition, which can detect both α-synuclein and Aβ aggregates[54]. Most of the aggregates were smaller than 100 nm, with control samples having a higher proportion of larger aggregates (Fig. 7A–D). The length

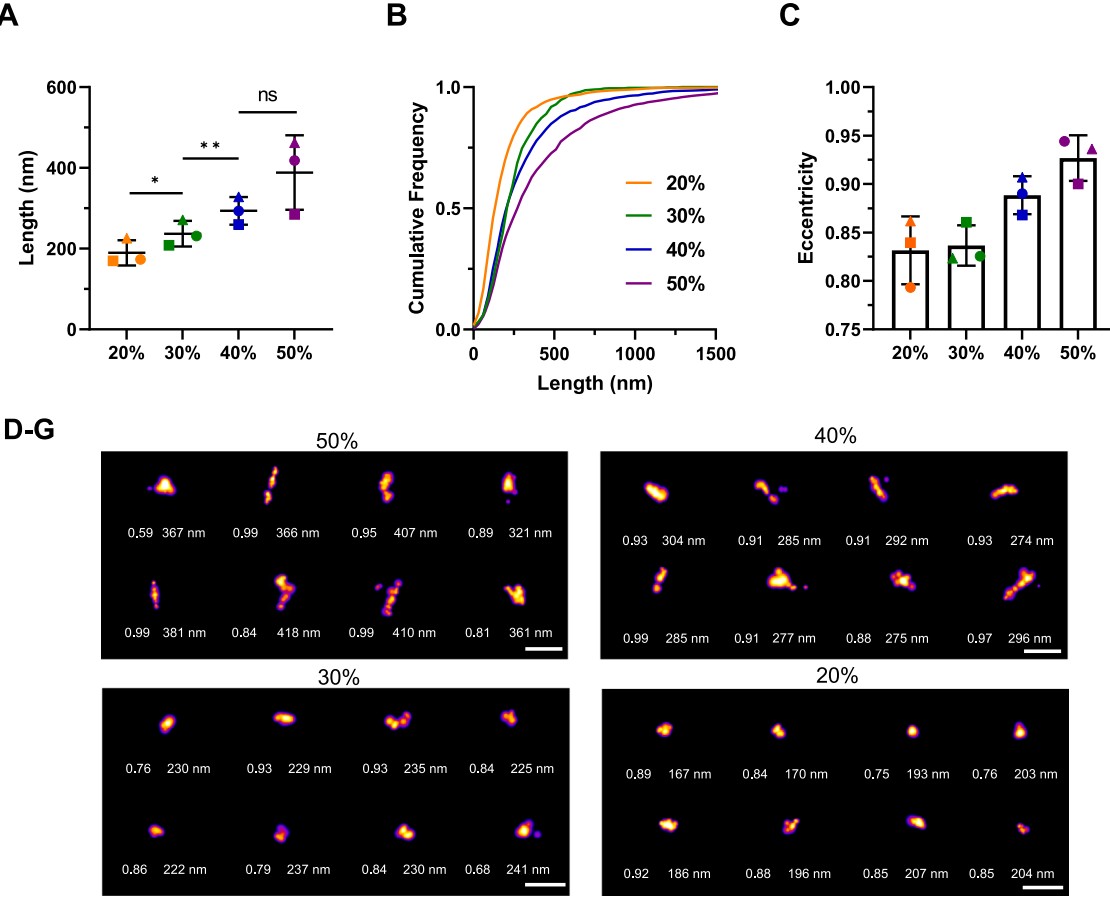

**Fig. 3 | Length distribution of the sucrose fractions from single-molecule super-resolution microscopy using thioflavin X. A** Scatter plot with each point representing one of three biological replicates. Error bars are mean ± STD. Statistical testing using two-tailed paired *t*-test from N = 3, *p* = 0.0144 for 20% vs 30%, *p* = 0.0496 for 30% vs 40%, *p* = 0.2939 for 40% vs 50%. **B** Cumulative frequency distribution of α-synuclein aggregates in the correspondent sucrose fractions. One representative out of three independent replicates. **C** Mean eccentricity for each fraction with each point representing one of three replicates. Error bars are mean ± STD from three independent replicates. **D–G** Representative super-resolution images from each fraction including their length and eccentricity. The LUT Fire has been applied to all images. Scale bar = 500 nm. Selected representatives out of all three replicates displayed here. Each individual colour is representing a distinct sucrose fraction (orange = 20%, green = 30%, blue = 40%, violet = 50%). Source data are provided as a Source Data file.

distribution of the aggregates differed between PD and control brain extracts with a median aggregate length of 55 nm for PD and 68 nm for controls (Fig. 7E). Length distributions plotted for individual brain samples are shown in Supplementary Fig. 6. We plotted the relative differences between PD and controls for the normalized and cumulative histograms of the length distributions (Fig. 7B, D). Negative values indicate that more species of that particular length are present in the control group. There were about 3% more aggregates smaller than 50 nm in the PD samples, whereas aggregates larger than 100 nm in length are 1% more abundant in the controls (Fig. 7B). According to the cumulative length distributions (Fig. 7D), there was the biggest difference (-13%) in the number of aggregates smaller than 74 nm in PD versus controls. Representative super-resolution images are shown for PD (Fig. 7G) and controls (Fig. 7H). The PD and control aggregates had the same eccentricity of 0.78 indicating that they are predominantly non-fibrillar (Fig. 7D). Overall, these data show that smaller non-fibrillar aggregates <100 nm are present in the PD brain samples over the control samples.

To independently confirm these results, we performed high-resolution AFM phase-controlled imaging on the smaller soluble aggregates from the brain samples (Supplementary Fig. 7). In this case, due to the increased complexity of the samples it was not possible to resolve monomers, so the AFM and super-resolution data are in closer agreement. The results confirm that PD

soluble aggregates had slightly smaller median height (PD 4.2 ± 0.05 nm versus control 4.4 ± 0.05 nm, *p* < 0.001) and smaller cross-sectional diameter than control ones (PD 20 ± 1 nm versus control 22 ± 1 nm, *p* < 0.001). Consequently, the AFM data confirmed the super-resolution results that a higher proportion of smaller non-fibrillar aggregates are present in the PD samples compared to the controls.

Since the aptamer cannot distinguish between α-synuclein and Aβ, we further used an antibody-based approach in order to understand the composition of the extracted aggregates. We used the SiM-Pull method described above[42] to compare aggregates derived from the PD and control brain samples. We chose the 6E10 antibody to detect Aβ and SC for detecting the toxic α-synuclein aggregates. The SC antibody was chosen since it detected the most aggregates in the 10 and 20% sucrose fractions, which exhibited the highest toxic response in our biological assays (Figs. 5, 6). We compared the fluorescence intensity for α-synuclein and Aβ between PD and control samples (Fig. 8A and Supplementary Fig. 8B). By using the same pair of antibodies for capturing and detection, we avoid the detection of monomeric aggregates since they only have one accessible epitope. In the case of 6E10 (Supplementary Fig. 8), there was no significant difference in the number of spots or the intensity between control and PD brain-derived aggregates, suggesting that the differences in aggregate length seen with AD-PAINT are not driven by Aβ. When using the SC

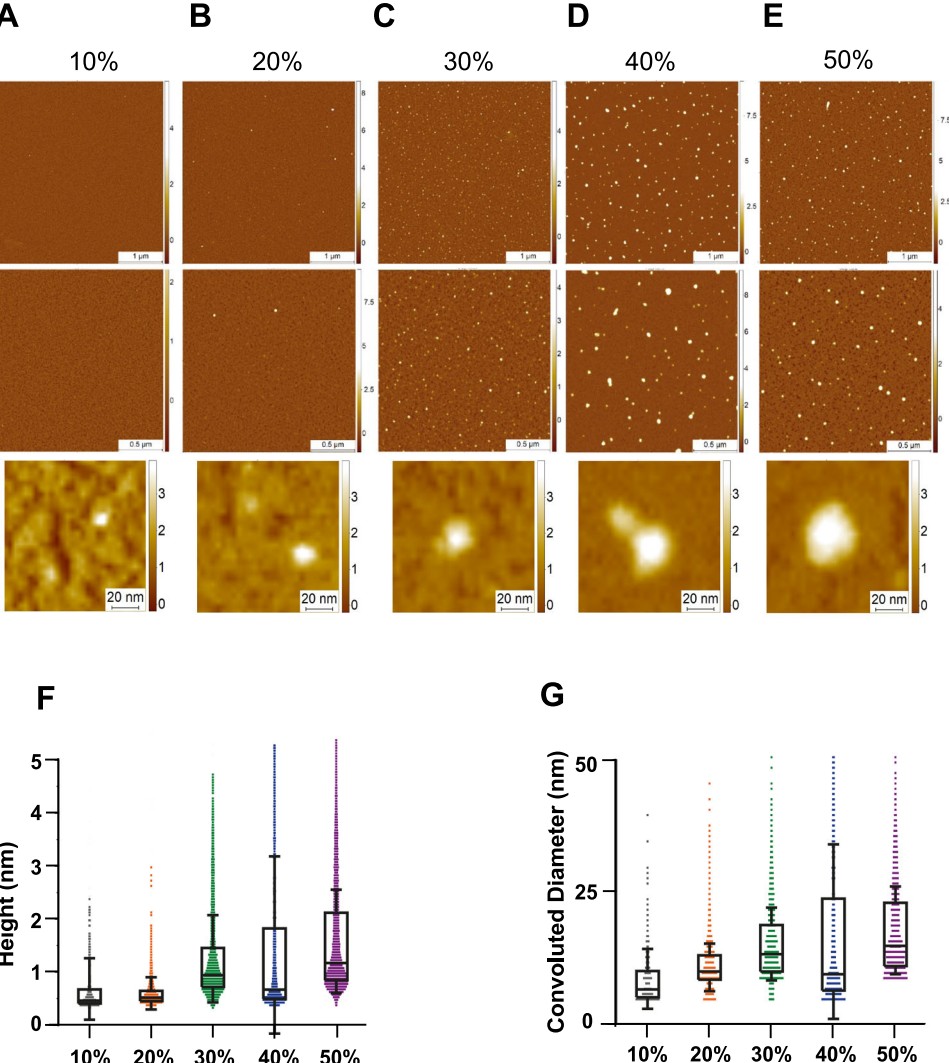

**Fig. 4 | Characterization of 3D morphology of different sucrose fractions using high-resolution AFM imaging. A–E** Representative AFM images of the sucrose fractions including zoom-in snapshots of individual aggregates. **F, G** Boxplot of the height ($N = 527$ for 10%, $N = 2785$ for 20%, $N = 6884$ for 30%, $N = 2473$ for 40% and $N = 7333$ for 50%) and convoluted diameter ($N = 533$ for 10%, $N = 2786$ for 20%, $N = 6885$ for 30%, $N = 2445$ for 40% and $N = 6585$ for 50%) (**G**) of each fraction. The limits of the box representing the 25th–75th quartile range and the horizontal line matching the median and the whiskers are mean + 1 STD. Each individual colour is representing a distinct sucrose fraction (orange = 20%, green = 30%, blue = 40%, violet = 50%). The datasets represents one technical repeat. Source data are provided as a Source Data file.

antibody (Fig. 8A–C), the detected α-synuclein aggregates derived from PD brains showed no difference in total number of detected aggregates but were lower in intensity than those from control brain (mean PD 35863 vs HC 40477 AU). Aggregate brightness correlates with size with smaller aggregates appearing less bright. The maximum cumulative difference in intensity was 6% between PD and controls, suggesting that the differences in α-synuclein aggregates represent the size discrepancy between PD and controls observed using AD-PAINT imaging. To confirm that the control group consists of longer aggregates, direct STORM-SiMPull was performed with the aggregate-specific MJF antibody (Supplementary Fig. 9A), which showed the highest binding towards fibrils using the α-synuclein sucrose fractions (Fig. 5B). The cumulative frequency distribution shows that the PD group have more aggregates smaller than 80 nm and the controls having longer aggregates exceeding 100 nm in length. Taking the ratio between detected spots using the MJF and the SC antibody (Supplementary Fig. 9B) revealed that the PD brain extracts contain less fibrillar α-synuclein aggregates, detected using the MJF antibody, than the control group. Hence PD brain extracts contain smaller aggregates

than the control brain extracts, in agreement with the AFM and super-resolution imaging data (Fig. 7 and Supplementary Fig. 7).

Aggregates extracted from PD patients, which comprise smaller α-synuclein aggregates compared to controls (Figs. 7, 8 and Supplementary Figs. 7, 9), also produce a greater inflammatory response when cultured with BV2 mouse microglial cells over a time course of 96 h (Fig. 8E). The data were normalized to the respective protein concentration. Since the total quantity of aggregates determined by AD-PAINT (Fig. 8D) is similar, this difference in inflammatory capacity is presumed to be related to the difference in α-synuclein aggregate size, and not driven by the number of aggregates. The BV2 cells were also cultured with lipopolysaccharide (LPS) as a positive control, which caused a high TNFα response and with buffer only as a negative control (B) (Supplementary Fig. 10). The buffer did not show any significant difference compared to untreated cells (UNT). The membrane permeabilization assay was not sensitive enough to detect responses from the soaked brain-derived samples, possibly due to the lower abundance of aggregates in comparison to the recombinant α-synuclein preparations.

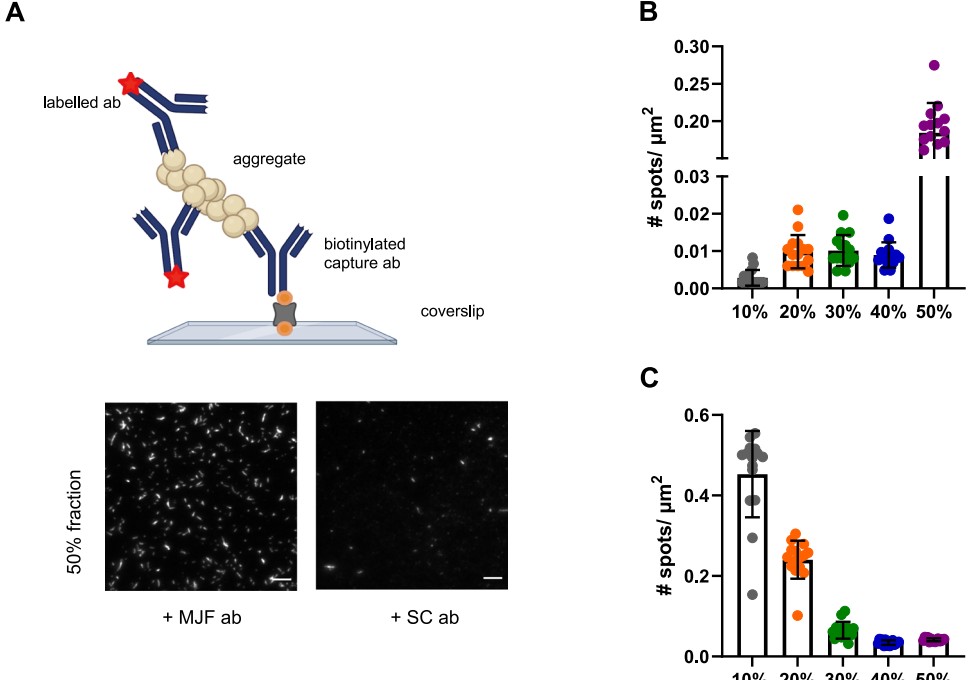

**Fig. 5 | Different α-synuclein species are present in different sucrose fractions.**
**A** Principle of the single-molecule pulldown (SiMPull) assay using the same pair of antibodies (abbreviated as ab in the schematic), including representative TIRF images for the 50% fraction using the MJF and SC antibodies. The biotinylated capture antibody (abbreviated as ab in the schematic) is attached to the passivated surface via a biotin (orange) neutravidin linkage (grey). The capture antibody can pulldown its distinct antigen. The captured aggregates are sandwiched by primary monoclonal fluorescently labelled (red star) detection antibody. Scale bar = 5 μm. The images are contrast adjusted. The schematic has been created with BioRender.com. **B** Number of detected species in each sucrose fraction using the conformation-specific MJF antibody targeting aggregated filamentous α-synuclein.

Error bars are mean ± STD from 15 different fields of view. Each individual point is representing one field of view. **C** Number of detected aggregates in each species using the sequence-specific SC epitope-specific antibody targeting amino acids 121–125. Error bars are mean ± STD from 15 different fields of view. Each individual point is representing one field of view. One representative replicate out of three is displayed here. Each individual colour is representing a distinct sucrose fraction (grey = 10%, orange = 20%, green = 30%, blue = 40%, violet = 50%). Source data are provided as a Source Data file. MJF = MJFR-14-6-4-2 antibody, SC = Santa cCruz 211 antibody, μm = Micrometer, # = Number. Details about the antibodies can be found in Supplementary Table 3.

## Discussion

Since α-synuclein aggregates are highly heterogeneous, it is difficult to determine how the physical characteristics such as size and morphology determine their neurotoxic properties. Different hypotheses have been suggested about which exact aggregate forms lead to cell death. While initial studies suggested that amyloid fibrils are the primary neurotoxic species involved in neuronal death due to their abundance in Lewy bodies[55,56], the focus has shifted in recent years towards the soluble α-synuclein oligomers, which form on the pathway of fibril formation. The challenges in characterizing these oligomeric species are that they are highly heterogeneous and less abundant compared to the monomer species[11,14–16]. In contrast to previous studies that have used mutant forms of α-synuclein or protocols to enrich oligomers in certain states[11,14,16,32,34–39], we have used here wild-type α-synuclein and studied the wide range of species of different size and structure formed during an aggregation reaction.

Using different ultrasensitive biophysical methods, we were able to characterize differently sized α-synuclein aggregates. To separate the aggregates in size, we chose a centrifugation-based technique, which is shown to be non-perturbative and reproducible[40,41]. By using the sucrose density gradient separation, we were able to isolate five different fractions, which all exhibit different species lengths and eccentricities as shown with ThX super-resolution microscopy, ThT imaging, TEM and AFM and SiMPull. Each of these methods has its advantages and limitations. ThT and ThX are both β-sheet-specific dyes, which makes it difficult to study species lacking β-sheet structures. We have seen that the content of β-sheet containing aggregates correlates with higher sucrose densities confirming that the

aggregates contain higher amounts of β-sheet structures with increasing fractions. The resolution of the ThX super-resolution images is about 30 nm and the images obtained are of the aggregates in solution[44].

TEM and AFM are label-free high-resolution methods, which rely on the drying of the sample on grids or mica. Both methods revealed the presence of small, pre-fibrillar spherical aggregates in the 10 and 20% fractions. The 30% fraction contained elongated aggregates. However, TEM only returns 2D projections of the aggregates, while AFM allows analysis of their 3D morphology and volume. According to the AFM analysis, the 20 and 30% fractions contained α-synuclein protofilaments, whereas the 40 and 50% consisted mainly of larger species. The size of the fractionated proteins increases as a function of sucrose gradient as confirmed independently by single-molecule fluorescence and AFM. It must be noted that these techniques are complementary and should not be expected to return identical values. Average values in diameter of the fractionated aggregates measured by these two techniques are different due to their intrinsic sensitivity and the use of different surfaces for deposition. The affinity to small and large aggregates varies, which causes higher or lower average values of the diameter of sampled species compared to the distribution of the populations in solution. When measuring average of particle sizes in number, AFM is largely biased towards small particles as the technique is sensitive towards monomers and small oligomers. When measuring on negative mica surfaces at a scale of 4 × 4 μm, it is also less sensitive in number to larger aggregates (>100 nm). Conversely, single-molecule fluorescence is insensitive to single monomers and small non-β-sheet aggregates which are not ThX active and

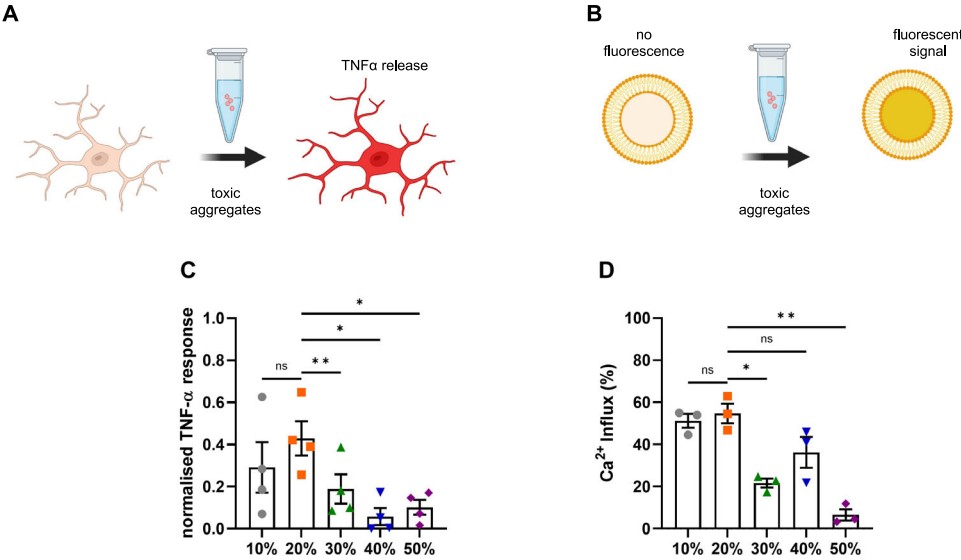

**Fig. 6 | Size is linked to toxicity. A** Upon addition of toxic aggregates, BV2 microglial cells release TNF-α into the culture medium, which is measured in the supernatant. **B** Liposomes are filled with a $Ca^{2+}$-sensitive dye. Upon incubation with toxic aggregates, $Ca^{2+}$ from the buffer can enter the liposome and cause a fluorescent signal. **C** TNF-α response from BV2 cells upon 24 h incubation with different sucrose fractions. The data are normalized to the negative control (buffer) and positive control (LPS at 10 ng/mL). Error bars are mean ± SEM from four independent biological replicates. Matched one-way ANOVA with Geisser-Greenhouse correction and Dunnett's multiple comparison test with a significance level of $\alpha = 0.05$. Corresponding $p$-values are 10% vs 20% $p = 0.288$, 20% vs 30% $p = 0.006$,

20% vs 40% $p = 0.020$ and 20% vs 50% $p = 0.035$. **D** $Ca^{2+}$-influx for aggregates from each fraction measured by membrane permeability assay. Error bars are mean ± SEM from three independent biological replicates. Matched one-way ANOVA with Geisser-Greenhouse correction and Dunnett's multiple comparison test with a significance level of $\alpha = 0.05$. Corresponding $p$-values are 10% vs 20% $p = 0.899$, 20% vs 30% $p = 0.041$, 20% vs 40% $p = 0.142$ and 20% vs 50% $p = 0.008$. Each individual colour is representing a distinct sucrose fraction (gray = 10%, orange = 20%, green = 30%, blue = 40%, violet = 50%). The schematics have been created with BioRender.com. Source data are provided as a Source Data file.

**Table 1 | Brain donor demographic information**

| Diagnosis | Sex | Age (years) | Disease duration (years) | Postmortem interval (hours) | Cause of death |
|---|---|---|---|---|---|
| Control | M | 72 | – | 55 | Myocardial infarction |
| Control | M | 70 | – | 12 | Carcinoma of the oesophagus |
| Control | M | 87 | – | 61 | Ruptured aortic aneurysm |
| Parkinson's disease | M | 86 | 18 | 6 | Parkinson's disease |
| Parkinson's disease | M | 64 | 19 | 108 | Congestive cardiac failure |
| Parkinson's disease | M | 75 | 12 | 51 | Parkinson's disease |

capable of measuring the size of very large aggregates at a scale of $200 \times 200\ \mu m$ skewing the average to larger values.

With ThX super-resolution, we measured the length and eccentricity from the SR images but lacked structural insight. Then, to gain further structural information, we used an antibody-based imaging approach. The MJF antibody, which binds to aggregated/filamentous α-synuclein, detected roughly 20 times more aggregates in the 50% high-density fractions compared to the lower-density fractions. In contrast, the SC antibody, which targets amino acid residues in the C-terminus of α-synuclein, detected 10 times more aggregates in the lower sucrose density fractions, which have no large fibrillary species present. This is in agreement with previous structural studies of α-synuclein aggregates, indicating that the C-terminal domain is not accessible in fibrils[50].

After assessing their structural properties in great detail, we used two different biological assays as a measure of toxicity, normalizing to total protein concentration to allow direct comparison between the fractions. We first studied their ability to disrupt lipid membranes, and we observed that the smaller α-synuclein aggregates predominantly found in the 10% and 20% sucrose fractions were more potent in causing membrane destabilisation and permeabilization. We then confirmed that the small aggregates in these same fractions lead to an

increased release of the pro-inflammatory cytokine TNFα from BV2 microglial cells. Our finding that smaller α-synuclein aggregates lead to a greater inflammatory response is in agreement with previously published work, which has shown that small α-synuclein oligomers are more likely to induce TNFα production in BV2 cells compared to large α-synuclein fibrils[27]. There has been extensive study on the size-dependent properties of Aβ[40], revealing that the smaller aggregates are more prone to cause membrane permeabilization and inflammation, in agreement with our findings for α-synuclein and suggesting that this might be a general property of aggregates.

Taking all of these results together, we isolated five differently sized α-synuclein species and observed that with increasing sucrose density, the length, eccentricity, and abundance of β-sheet containing aggregates increased and that the aggregates with accessible C-terminus decreased. We correlated aggregate size with toxicity and found that small aggregates, <200 nm in size, were the most toxic species in our biological assays. This also allows us to predict the toxicity of samples, whose concentration is too low to perform toxicity measurements, based on the size of the aggregates present.

So far, soluble Aβ aggregates isolated from AD brains were shown to cause long-term potentiation deficits but comparable studies using PD post-mortem tissue were missing[57]. In this initial work, we have

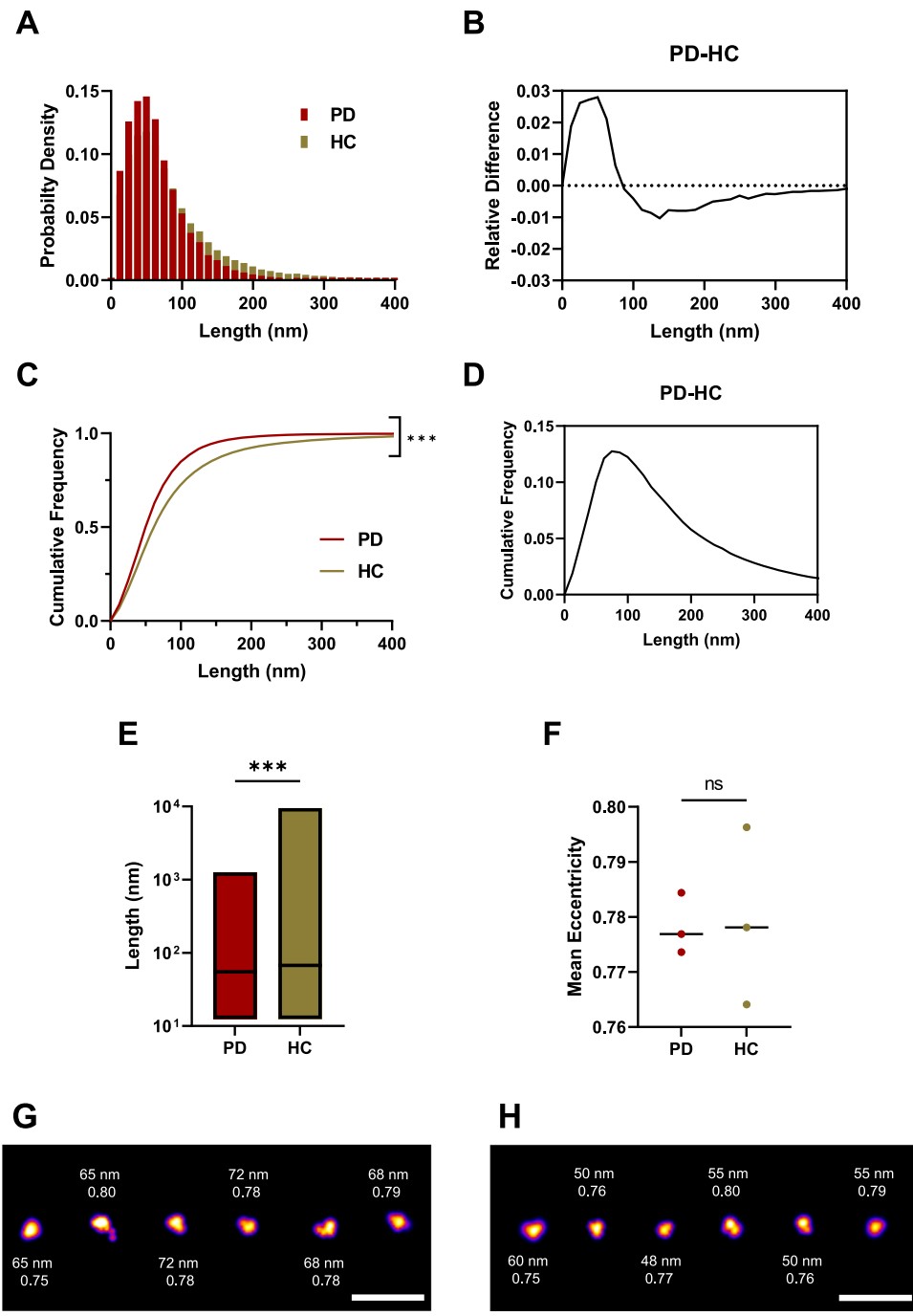

**Fig. 7 | The soluble aggregates extracted from PD and control brains have different sizes as super-resolved with AD-PAINT. A** Histogram of the size distribution for PD (burgundy, $N = 54218$) and control brain-derived aggregates (beige, $N = 48222$). Each histogram is the collected data from three cases. **B** Difference between the histograms for size distribution. **C** Cumulative frequency distribution for aggregate length for PD and controls. Each histogram is the collected data from three cases (Kolmogorov-Smirnov test, $p < 0.001$ for PD versus controls). **D** Difference between the cumulative frequencies for the size distribution. **E** Boxplot comparing aggregate length in PD versus controls. Box limits represent minimum and maximum value, horizontal line in the centre indicating median (Mann–Whitney $U$ test, $p < 0.001$). **F** Scatter plot of the mean eccentricity of all aggregates with each point representing the mean value of three patients per group (two-tailed unpaired $t$-test, $p = 0.909$, $N = 3$ for PD and controls). **G** SR representative images for PD cases showing the corresponding aggregate size and eccentricity. **H** SR representatives for controls showing the corresponding aggregate size and eccentricity. The LUT Fire has been applied to all images. Scale bar = 500 nm. Source data are provided as a Source Data file. PD = Parkinson's disease patients vs HC = control patients.

extracted soluble aggregates from post-mortem brain tissue from three PD patients and three controls and analysed the length, composition and toxic properties of the extracted aggregates. Insoluble α-synuclein aggregates are known to be present in PD brains but not in control brains[58], but these are not extracted in the soaking process. Soaking leads to the extraction of only soluble aggregates which were present at similar concentrations in the PD and control brains. This avoids any potential artefacts due to homogenisation of large insoluble aggregates. Previous work has shown that soaking is a minimally perturbative method to preferentially extract the toxic aggregates from post-mortem brain, producing samples containing aggregates of the size and structure found in vivo[53]. Interestingly, the soluble

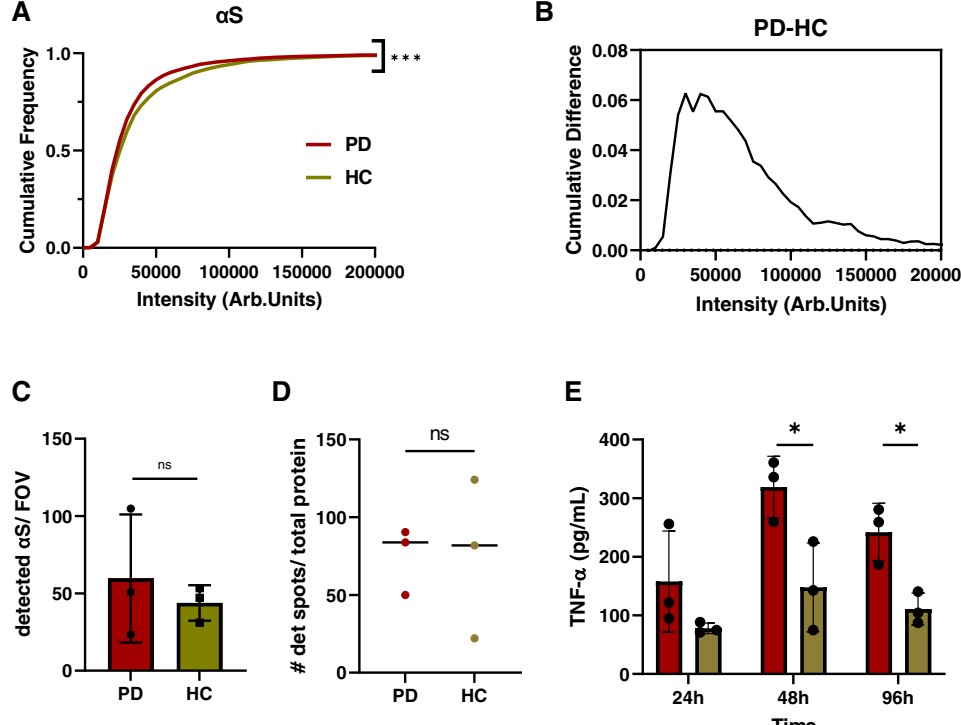

**Fig. 8 | Smaller aggregates in PD brains comprise α-synuclein and PD-derived aggregates induce a greater inflammatory response. A** Cumulative frequency distribution for α-synuclein aggregate intensity for PD (burgundy, $N = 6866$) and control (beige, $N = 3157$) measured using SiMPull with the epitope-specific SC antibody. Each histogram is the collected data from three cases. (Kolmogorov-Smirnoff, $p < 0.001$ for PD vs controls). **B** Cumulative difference for α-synuclein intensity for PD and controls, measured using SiMPull. **C** Number of detected α-synuclein aggregates per field of view for PD ($N = 3$) and controls ($N = 3$). Error bars are mean ± STD from three cases. (two-tailed unpaired $t$-test, $p = 0.559$, $N = 3$). Each point represents one of the three patients per group. **D** Scatter plot for the number of detected spots per field of view determined by AD-PAINT. Each point represents the mean value from different replicates for each of the three PD cases and controls and the line represents the median (two-tailed unpaired $t$-test from $N = 3$ per group, $p = 0.971$). The data are normalized to the total protein concentration determined by BCA. **E** TNF-α response from BV2 cells upon incubation with brain extracts over a distinct time course. The data are normalized to the total protein content determined by BCA. Error bars are mean ± STD from three patients per group. (two-tailed unpaired $t$-test, for 48 h $p = 0.033$ and for 96 h $p = 0.016$, $N = 3$ for PD and controls). Source data are provided as a Source Data file. Arb.Units = arbitrary units, # = Number, PD = Parkinson's disease, HC = controls.

aggregates from the post-mortem brain were present in similar sizes and morphology to those formed by aggregating α-synuclein in a test-tube, but there were differences in the size distribution of the aggregates from PD patients and controls. This finding is different to what Je et al.[42] reported, possibly because they looked at both insoluble and soluble species. We found that in the PD brain extracts, the aggregates identified by AD-PAINT (both Aβ and α-synuclein) were shorter than the control extracts, and according to the SiMPull analysis, the α-synuclein aggregates are dimmer in the PD samples, which correlates with aggregate size. The PD samples were also more inflammatory compared to the control samples, suggesting that this difference is due to the presence of smaller α-synuclein aggregates in the PD samples. This is exactly what would be predicted based on the experiments using the different fractions of recombinant aggregates separated using the sucrose gradient, i.e. aggregates from controls, which are larger on average, would be less inflammatory. This result is similar to the result of our recent study on a Braak stage III AD brains, where we found that the hippocampus contained a higher proportion of smaller aggregates than the visual association cortex and that these hippocampal aggregates caused an increased inflammatory response[59]. There was closer agreement in the aggregate sizes measured by super-resolution imaging and AFM than for the sucrose gradient factions. This is because no fibrils were present and the more complex substrate background did not allow the AFM to detect monomeric and small oligomeric species. Both methods show that the largest difference in aggregate size for PD compared to controls is in the range of 0–100 nm. This refines the size range of the toxic species obtained

from the sucrose gradient suggesting it is the aggregates smaller than 100 nm that are most toxic.

The concentration of the aggregates in the soaked brain samples is low and probably close to the situation in vivo where low levels of aggregate-induced toxicity of long times lead to the development of disease. This means that sensitive assays are needed to measure the toxicity in these samples. Our highly sensitive membrane permeabilization assay was not sensitive enough to detect changes produced by the soaked brain samples. Therefore, we used an inflammatory assay but this assay still did not have a sufficient dynamic range to allow us to perform immunodepletions to establish if the inflammatory response was caused by Aβ or α-synuclein aggregates, although changes in the inflammatory response correlated with changes in the α-synuclein aggregates. A more sensitive inflammatory assay would be needed to make these experiments feasible. In addition, it would be beneficial to develop sensitive toxicity assays for neurons that work at the low levels of aggregates present in these samples and explore if differences in the α-synuclein aggregate heterogeneity or toxicity correlate with patient symptoms or pathology.

Based on our data, we propose that neurons have mechanisms to form and secrete larger, less toxic soluble aggregates as a route of aggregate removal, thereby preventing neurotoxicity. These mechanisms presumably fail during the development of PD, leading to the release of more small toxic aggregates as well as the formation of insoluble aggregates, Lewy bodies, in some neurons. Future work will explore the variation in the number and size of aggregates in different brain regions and be performed on a larger number of brain samples.

Understanding what factors control the size and structure of the aggregates produced in the brain is clearly important to understand what cellular processes fail and lead to the development of PD, and may also be related to the patient symptoms.

In summary, we have shown that the small soluble non-fibrillar α-synuclein aggregates smaller than 200 nm in length are highly toxic species causing lipid membrane permeabilization and inflammation. Furthermore, in human post-mortem samples from the amygdala, a critical brain region where Lewy body pathology and neuroinflammation are observed in PD, the soluble aggregates are significantly smaller and more inflammatory than controls. Hence, our study provides evidence that these small soluble aggregates are the critical species driving toxicity in the PD brain.

## Methods

### α-synuclein aggregation and separation by density centrifugation

Monomeric α-synuclein was expressed and purified from *Escherichia coli* as described previously[60]. The initial stock was diluted to 1 mg/mL in prefiltered sodium phosphate buffer (pH 7.4) (0.02 μm syringe filter, Whatman, 6809-1102) and afterwards ultracentrifuged at $350,000 \times g$ for one hour at 4 °C using a TL120.2 rotor (Beckman). For the aggregation reaction, the tube was placed into a 37 °C incubator during constant shaking (200 rpm). Out of this tube, aliquots were taken after 0, 3, 6, 9, 16, 24 and 48 h. These time points were layered on a sucrose step gradient with discontinuous densities (10–50% w/v) using 400 μL of sucrose solution per layer.

The sucrose gradient was centrifuged for 4 h at $113,000 \times g$ at 4 °C in a Beckman ultracentrifuge using a SW 60 Ti swinging-bucket rotor. Afterwards each fraction was collected starting from the top to the bottom by piercing the tube orthogonally with a needle. In order to remove the sucrose from the fractions, dialysis with three buffer exchanges in sodium phosphate-containing buffer using Slide-A-Lyzer MINI Dialysis Devices (3.5 K MWCO, Thermo Scientific, Cat. 69550) was performed. The dialysed fractions were aliquoted and stored at −80 °C and then thawed and analysed when needed.

### Extracting soluble aggregates from brain tissue

Post-mortem brain tissue from three PD cases and three age and sex-matched controls with no known history of neurological or neuropsychiatric symptoms was acquired from the Cambridge Brain Bank (with the approval of the London−Bloomsbury Research Ethics Committee; 16/LO/0508, Table 1). The brain samples have been voluntarily donated without any compensation. Brains were flash-frozen and stored at −80 °C at the John van Geest Centre for Brain Repair in Cambridge. For extraction of soluble aggregates, a previous published protocol was adapted[53]. 300 mg was cut from the amygdala and placed into an eppendorf containing 1.5 mL of artificial cerebrospinal fluid buffer (aCSF, 124 mM NaCl, 2.8 mM KCl, 1.25 mM $NaH_2PO_4$, 26 mM $NaHCO_3$; pH 7.4, supplemented with 5 mM EDTA, 1 mM EGTA, 5 μg/mL leupeptin, 5 μg/mL aprotinin, 2 μg/mL pepstatin, 20 μg/mL Pefabloc, 5 mM NaF) for 30 min at 4 °C. Afterwards the samples were centrifuged at $2000 \times g$ for 10 min and 90% of the supernatant was transferred into a fresh tube. This solution was then centrifuged at $14,000 \times g$ for 2 h. The upper 90% supernatant was collected and dialysed for 72 h using Slide-A-Lyzer cassettes (MKCO 2 kDa, Thermo Scientific, Cat. 66330) with three buffer exchanges against aCSF buffer at 4 °C. The samples were aliquoted and stored at −80 °C and each aliquot was used for just one experiment in order to avoid unnecessary freezing/thawing cycles.

### Immunohistochemistry on human brain tissue

Immunohistochemistry was done on 10 μm-thick formalin-fixed, paraffin-embedded postmortem brain sections of the amygdala. Brain sections were rehydrated sequentially in xylene, 100% EtOH, 90% EtOH, 70% EtOH and dH₂O. Antigen retrieval was performed with 98%

formic acid (pH = 1.6–2.0) for 5 min. Endogenous peroxidase activity was blocked with 4% $H_2O_2$ in dH₂O, for 10 min. The sections were then washed 3× in PBS and blocked with 2% milk in PBS for 30 min at room temperature. Sections were subsequently incubated with an alpha-synuclein antibody (Enzo Life Sciences sa3400, 1:250 dilution) in 2% milk for 1 h at room temperature. Sections were then rinsed 3× times with PBS and incubated with 1% secondary antisera/HRP conjugate in 10% normal human serum in PBS for 1 h. After 3× rinses with PBS, the colour was developed by adding DAB solution (Vector Laboratories). Upon rinsing sections with dH₂O, nuclei were counterstained with Haematoxylin (Thermo Scientific) for 30 s, followed by blueing in tap water for 5 min. Finally, sections were sequentially dehydrated in ascending EtOH concentrations and coverslipped using DPX mounting medium. Slide scanning was done at the Histopathology/HIS facility at the Cancer Research UK Cambridge Institute. Scanning was performed on the Aperio Scanscope AT2 (Leica Biosystems) at ×20 magnification with a resolution of 0.503 μm per pixel. Images were viewed with the Aperio Imagescope viewing platform (Leica Biosystems).

### Protein concentration estimation

After dialysis, the concentration of the sucrose samples was estimated with Bradford Assay (Thermo Scientific, Cat. 23236) and the concentration of the soaked brain with BCA (Thermo Scientific, Cat. 23225) in a 96-well microplate (Supplementary Table 2). Triplicates of a known bovine serum albumin sample (standard) and samples of interest were pipetted into a microplate in equal volumes. Afterwards the Coomassie G-250 dye in the case of the Bradford assay or BCA reagent A and B mixture was added to each well and the plate was incubated at RT for ~30 min. The absorbance was measured at 595 nm for the Bradford assay and at 562 nm for the BCA assay in a multiplate reader (Clariostar).

### Neuroinflammatory assay

BV2 cells mouse microglia cells obtained from ICLC and cultured in T75 flasks in Dulbecco's Modified Eagle Medium (Gibco, Life Technologies, Cat. 21063-029) containing 10% foetal bovine serum (Sigma-Aldrich, St. Louis, MO, Cat. F0926), 1% penicillin streptomycin (Gibco, Life Technologies, Cat. 15140-122) and 1% L-glutamine (Gibco, Life Technologies, Cat. 25030-024) at 37 °C with 5% $CO_2$. For the assay, either 330,000 cell/mL for a 24 h experiment or 165,000 cell/mL for a 96 h were plated into flat-bottom 96-well plate (Gibco, Life Technologies, Cat. 25030-024). The fractions were diluted to a final concentration of 500 nM in the above mentioned media with 1% FBS and added to the cells for 24 h. Aggregates extracted from brain tissue were added to the cells in a 1/5 dilution and every 24 h the medium was changed for four consecutive days. All samples were processed using at least three different wells in order to account for variability. The next day, the supernatant was collected and stored at −80 °C.

For TNF-α quantification the mouse TNF-α -ELISA kit (R&D Systems, MN, USA, Cat. DY410) was used according to the manufacturer's protocol. Absorbance was measured at 450 nm in a plate reader (Clariostar). The results were normalized according to the total protein concentration determined via BCA.

### Membrane permeability assay

Membrane permeability was determined by the quantification of $Ca^{2+}$ influx induced in the presence of each sucrose fraction as described previously[61]. Briefly, liposome vesicles made of 1-palmitoyl-2-oleoyl-sn-glycero-3phosphocholine (POPC) containing 100 μM Cal-520 dye were tethered to a PLL-PEG coated coverslip (VWR International, 22 × 22 mm, product number 63 1-0122). Identical positions were imaged (3 × 3) under different conditions (background, addition of α-synuclein fractions and positive control with ionomycin (Cambridge Bioscience Ltd, Cambridge, UK)) on a home-built total internal reflection setup. For excitation a 488 nm laser (10 W/cm²) was used.

The acquired images were averaged over 50 frames with an exposure time of 50 ms. For analysis, the average Ca2+ influx per field of view was calculated according to Eq. 1:

$$Ca^{2+} = \left( \frac{F_{sample} - F_{background}}{F_{ionomycin} - F_{background}} \right) * 100\% \qquad (1)$$

where $F$ is fluorescence intensity of each spot under the three different conditions, namely blackground ($F_{background}$), in the presence of the sample ($F_{sample}$), and after the addition of ionomycin ($F_{Ionomycin}$).

## Preparation of thioflavin T and thioflavin X

Thioflavin T (ThT) and thioflavin X (ThX) stock solutions were prepared by dissolving ThX/ThT (Sigma-Aldrich) into neat dimethyl sulfoxide (DMSO, Sigma Aldrich, 276855). Afterwards the solution was diluted into prefiltered PBS (0.02 μm syringe filter, Whatman, 6809-1102) until a concentration of 100 μM was reached. The concentration was determined with a spectrophotometer (Nanodrop, molar extinction coefficient 30538 for ThT at $\lambda$ = 413 nm and 31,130 for ThX at $\lambda$ = 420 nm). ThT was imaged at a final concentration of 5 μM and ThX at 500 nM. Both dyes were stored at 4 °C upon usage

## Preparation of aptamer and imaging strand

The imaging strand (CCAGATGTAT-Cy3b) and the docking strand-conjugated aptamer (GCCTGTGGTGTTGGGGCGGGTGCGTTATAC ATCTA) were obtained from ATDBio. They were both diluted in PBS to a final concentration of 2 and 100 nM.

## Coverslip preparation for imaging

Before imaging, glass coverslips (20 × 20 mm, VWR International, 631-1570 or 24 × 50 mm, VWR International, 631-0146P) were placed in an argon plasma cleaner (PDC-002, Harrick Plasma) for 1 h. Incubation chambers (9 × 9 mm, Bio-Rad Laboratories Ltd, SLF-0201 for 20 × 20 mm coverslips and 50 wells, 1 × 3 mm, Merck, GBL-103250-10EA for 24 × 50 mm coverslips) were attached on the coverslip and either coated with 50 μL of poly-l- lysine (PLL, Sigma, 1 mg/mL) for 30 min for ThT/ThX imaging or 1% tween20 (Sigma-Aldrich, Cat. P1379-25ML, diluted in PBS) for DNA-PAINT experiments, followed by three washing steps with PBS (pH 7.4). For super-resolution imaging, an additional step of covering the surface with TetraSpeck Microsphere beads (0.1 μm, Invitrogen, Cat. T7279) was required for drift correction, followed by three washing steps with PBS. The samples were incubated for 5 min for in vitro aggregates and 1 h for soaked brain samples, afterwards removed and replaced with the dye of interest.

## Diffraction-limited total internal reflection fluorescence (TIRF) imaging

Image acquisition was performed on a bespoke microscope using an Eclipse TE2000-U body (Nikon Corporation) and a perfect focus system. ThT imaging was done using a 405 nm laser (LBX-405-50-CIR-PP, Oxxius) for excitation. The laser passed through a quarter wave plate (WPQ05M-405, Thorlabs) and was coupled into a 60× Plan Apo TIRF, NA 1.45 oil-immersion objective lens with an excitation power of 50 W/cm2. The fluorescence signal was collected by the same objective, separated from the excitation light by a dichroic (ZT 405/532 rpc), filtered by an emission filter (FF01-480/40/25) and focused onto an EMCCD camera (Evolve 512, Photometrics). Each pixel corresponded to 237 nm. The microscope was controlled with MicroManager.

## Single-molecule super-resolution imaging

Imaging using ThX was done with a 488 excitation laser (iBeam-SMART, Toptica) on a bespoke home-built total internal reflection microscope setup. For Aptamer DNA-PAINT imaging a 561 nm laser (Cobalt Jive, Cobalt) was used. The lasers were coupled into a 100x TIRF objective (NA 1.49, Apo TIRF, 60XO TIRF, Olympus) mounted on

an inverted Ti-E Eclipse microscope (Nikon, Japan). The microscope contains a Perfect Focus system counterbalancing z-drift. The fluorescence signal was collected by the same objective, separated from the excitation light by a dichroic (Di01-R405/488/561/635, Semrock) and subsequently passed through emission filters (LP02-568RS25, Semrock and FF01-587/35-25, Semrock). Afterwards the light was directed to an EMCCD-camera (Evolve 512, Photometrics) through a 1.5× beam expander. Each pixel corresponded to 98.8 nm. Data acquisition (6000–8000 frames, 50 ms exposure time) was carried out with MicroManager.

## Single-molecule pulldown assay (SiMPull)

Coverslip preparation was done as described previously[59,62,63]. Afterwards the coverslip was passivated with neutravidin (0.2 mg/mL) diluted in 0.05% PBS-T(v/v) (tween20 Cat. P1379-25ML diluted in prefiltered PBS) for 15 min. For Aβ capture, biotinylated 6E10 (BioLegend, Cat. 9340-02) antibody was diluted to 10 nM and for α-synuclein capture, 211 (Santa Cruz, Cat. sc-12767) or MFJR 14-6-4-2 (abcam, Cat. ab227047) at 10 nM were added and incubated for 10 min. The coverslip was washed with two cycles of 0.05% PBS-T(v/v) and one cycle of 1% PBS-T(v/v). The samples were incubated for 1 h; then the washing step was repeated. In order to minimise unspecific binding, a blocking solution containing 0.1% (w/v) bovine serum albumin (BSA) (Thermo Scientific, Cat. AM2616), 10% (v/v) salmon sperm (Thermo Scientific, Cat. 15632011) and 0.05% (v/v) PBST was added for 45 min. For detection, the Alexa-Fluor-647-labelled antibody corresponding to the biotinylated capture antibody was added for up to 30 min (6E10-647, Cat. 803021 at 500 pM, MJFR-14-6-4-2-647, Cat. ab216309 at 500 pM, 211-647, Cat. sc-12767 AF647 at 5 nM) followed by the washing step. Then 3 μL of PBS was added to each well and the coverslip was imaged. An overview about all the antibodies used in this work is given in Supplementary Table 3.

For direct STORM-SiMPull, three incubation chambers (Merck, GBL-103250-10EA) were stacked together using nail polish and were afterwards filled with 20 μL STORM buffer containing glucose oxidase (2 mg/mL, Sigma, G7141-250KU), catalase (52 μg/mL, Sigma, C3515) and cysteamine (7 mg/mL, Sigma, M9768-5G) at pH 8.0. The stacked imaging chamber was sealed with a coverslip to avoid oxygen penetration. The samples were excited on the above mentioned fluorescence microscope using a 638 nm laser (iBeam-smart, Toptica) at 150 mW with a constant 405 nm pulse (LBX-405-50-CIR-PP, Oxxius). 8000 frames at an exposure time of 30 ms were recorded.

## Biotinylation of antibody

DBCO-PEG4-biotin (Merck, Cat. No. 760749, Lot No. MKCN1219) was first dissolved in anhydrous DMSO at 10 mM as stock solution. It was then selectively conjugated on the carbohydrates of the Fc region of the monoclonal mouse anti-α-synuclein antibody (211) (Santa Cruz, Cat. No. sc-12767, Lot No. K3020) via a SiteClick™ Antibody Azido Modification Kit (Invitrogen, Cat. No. S20026) according to the manufacturer's instructions. In brief, 200 μg of antibody was concentrated and buffer exchanged in the provided antibody preparation buffer to 1.110 mg/mL by an Amicon spin filter (50 kDa MWCO). The antibody was then incubated overnight with β-galactosidase at 37 °C, followed by overnight coupling to UDP-GalNAz using β−1,4-galactosyltransferase (GalT) on the next day at 30 °C. The mixture was then purified by an Amicon spin filter (50 kDa MWCO, Merck, Cat. No. UFC505024). The concentration of the azido-modified antibody was calculated by A280 (2.229 mg/mL). With the azido-modified antibody, 10 molar equivalents of DBCO-PEG4-biotin was introduced for copper-free strain-promoted click reaction. After overnight incubation at 37 °C, excess DBCO-PEG4-biotin was removed by an Zeba™ Spin Desalting Column (40 kDa MWCO, ThermoFisher, Cat. No. 87766). The biotinylated antibody was then concentrated by an Amicon spin filter (100 kDa MWCO, Merck, Cat. No. UFC510024), and its concentration was

determined by A280 (0.727 mg/mL). The labelling efficiency was determined using Pierce™ Fluorescence Biotin Quantification Kit (ThermoFisher, Cat. No. 46610) (1.45 biotin/antibody).

## Transmission electron microscopy (TEM)

Sucrose fractions were added to glow-discharged, holey carbon grids (EM Resolutions, 400 mesh Copper) for 1 min, afterwards washed with distilled water twice for 30 s and lastly stained with uranyl acetate (1%, w/v) for 1 min. Images were acquired under minimal dose conditions with a Tecnai G2 transmission electron microscope at 200 kV.

## Atomic force microscopy (AFM)

Sucrose fractions of α-synuclein were deposited onto freshly cleaved mica which was positively functionalised using (3-Aminopropyl)triethoxysilane (APTES; 0.5% w/v), and incubated for 5 min before rinsing with 1 ml Milli-Q water. The samples were then dried using a gentle flow of nitrogen gas. AFM maps were acquired using NX10 AFM (Park Systems, South Korea) operating in non-contact mode. Imaging was performed in a constant phase change regime[46]. The set-up was equipped with a silicon nitride cantilever (PPP-NCHR) with a nominal tip radius of <10 nm and a spring constant of 5 N/m. Image processing was performed using SPIP software (Image Metrology, Denmark); images were first-order flattened and the height profiles measured. All measurements were performed at RT.

The diameter of each biomolecules is directly measured from the raw data without any further processing. AFM acquires 3-D maps of the morphology of biomolecules on a surface. By taking the cross-section along the Z direction with a line profiles of each biomolecule on the surface it is thus possible to measure their cross-sectional height and[45]. The measurements of height are precise at the angstroms level[40]. However, the measurement of the diameter of biomolecules of similar size to the AFM tip (<8–10 nm) are always subjected to an overestimation of the value, and called the convolution effect[64].

## Analysis and statistical testing

For diffraction-limited imaging, the frames were combined to a Z-Stack in imageJ (NIH, USA), cropped and contrast adjusted. Afterwards a threshold was chosen by comparing the number of spots to the negative control (PBS with dye). Only spots above this chosen threshold have been considered for analysis.

Super-resolution imaging was analysed using an open-source code publicly available on https://github.com/Eric-Kobayashi/SR_toolkit as previously described in ref. 54.

Briefly, the code is executing the function 'Peakfit' in the GDSC Single-Molecule Light Microscopy package (GDSCSMLM) plugin in imageJ. For the rendering of AD-PAINT (ThX) images, the two parameters 'signal strength' and 'precision' were set to 100 (60) and 20 (40). Later, DBSCAN (Python package scikit-learn) was used to cluster the individual localisations with a detection radius (epsilon) of 75 nm (200 nm) and a minimum localisation threshold of 9 (10). Images were corrected for XY-Drift by using the 'Drift Calculator' plugin within GDSCSMLM.

dSTORM data were analysed with the Nano plugin[65] for drift correction and the ThunderSTORM plugin[66] for super-resolution reconstruction using imageJ (NIH, USA).

Graphpad v9 was used for plotting the figures and statistical testing except for Fig. 4, which was plotted in origin. The datasets were checked for normal distribution and depending on the result, the appropriate statistical test was chosen. If two normally distributed datasets were compared, a t-test with a significance level of $\alpha = 0.05$ was used. If more than two datasets needed comparison a one-way analysis of variance (ANOVA) was chosen followed by a Dunnett's test when comparing the mean of each dataset to the mean of a control dataset. Kolmogorov–Smirnov was used to compare non-normally

distributed data distributions and Mann–Whitney for comparing ranks.

## Reporting summary

Further information on research design is available in the Nature Research Reporting Summary linked to this article.

## Data availability

Source data are provided with this paper. All other data are available from the corresponding author on request.

## Code availability

The code for rendering the super-resolution data is publicly available at https://github.com/Eric-Kobayashi/SR_toolkit and https://doi.org/10.5281/zenodo.4651484.

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

## Acknowledgements

Human post-mortem brain tissue was acquired from the Cambridge Brain Bank (Cambridge University Hospitals). The Cambridge Brain Bank is supported by the NIHR Cambridge Biomedical Research Centre. Immunostaining for α-synuclein, was performed by the Department of Histopathology and Cytology at the Cambridge Brain Bank. We thank Ewa Klimont and Swapan Preet for the expression and purification of α-synuclein. We thank Dr. Lisa Maria-Needham and Prof. Steven Lee for gifting thioflavin X. We would like to acknowledge James W.B. Fyfe, Dung T. Do and Prof. Thomas N. Snaddon for the synthesis of thioflavin X. We thank Dr. Wei Zhang and Prof. Ernst Laue for access to the ultracentrifuge. We thank Prof. Clare Bryant for the BV2 microglia cells. We would also like to acknowledge the Cambridge Advance Imaging Centre for their support in TEM imaging. Finally, we would like to thank all patients and their families for donating their brain to research, without you this work would have not been possible. This work was supported by Parkinson's UK (G-1901), UK Dementia Research Institute, which receives its funding from DRI Ltd. funded by the UK Medical Research Council and by the European Research Council Grant Number 669237 and the Royal Society. Y.Z. was supported by the UK Engineering and Physical Sciences Research Council (EP/R005397/1) and the NIHR Cambridge Biomedical Research Centre Dementia and Neurodegeneration Theme (146281). J.Y.L.L. is supported by the Croucher Foundation Limited (Hong Kong). A.K. was funded by the Onassis Foundation (Scholarship Program for Hellenes) and the Alborada Studentship from Wolfson College, Cambridge. D.I.S. is now working at AstraZeneca. C.H.W.G. was supported by a RCUK/UKRI Research Innovation Fellowship awarded by the Medical Research Council (MR/R007446/1), and the NIHR Cambridge Biomedical Research Centre Dementia and Neurodegeneration Theme (146281). The views expressed are those of the authors and not necessarily those of the NHS, the NIHR or the Department of Health. For the purpose of open access, the author has applied a Creative Commons Attribution (CC BY) licence to any Author Accepted Manuscript version arising from this submission.

## Author contributions

D.E., Y.P.Z., E.L., A.M., X.L., Z.X., H.D., D.I.S., J.Y.L.L., R.T.R, A.K., Y.Z., F.S.R. and S.D. performed and analysed the experiments. T.P.J.K., M.V., F.I.A., C.H.W.-G. and D.K. supervised the project. D.E. and D.K. conceived the idea, designed the study and wrote the manuscript. All authors discussed the results and proofread the manuscript.

## Competing interests

The authors declare no competing interests.
