## [Peer Review File · Nature Communications]

REVIEWER COMMENTS

Reviewer #1 (Remarks to the Author):

Drs. Emin and Klenerman's group showed that aggregates of wild-type α -synuclein (α syn) smaller than 200 nm in length, cause inflammation and permeabilization of liposome membranes. Studying soluble aggregates extracted from post-mortem human brains also reveals that these aggregates are similar in size and structure to the smaller aggregates. Furthermore, the soluble aggregates present in PD brains are smaller, largely less than 50 nm, and more inflammatory compared to the larger aggregates present in control brains. This study suggests that the small non-fibrillar α syn aggregates are the critical species in the pathogenesis of PD.

Major comments

1. How similar or different are the symptoms of the three PD patients? It would be nice to have an explanation as to whether the toxic aggregate sizes are similar for different symptoms.
2. Can the difference in length and shape between EM, AFM and observation using the patient's brain be explained only by the difference in the methods? What causes the difference of 50 nm and 200 nm in in vitro and actual brain systems.
3. It is better to describe how long it takes to measure the aggregate size after separation by sucrose density gradient centrifugation. If possible, it is better to describe the presence or absence of changes in size after separation.
4. Does the aggregate that has been structurally observed after separation cause cytotoxicity? The authors should examine whether all aggregates are still small in size after incubation time in the cell culture system.
5. The shape (linearity) of the aggregates contained in the fraction with high sucrose concentration appears to be different between the super-resolution fluorescence microscope (Fig. 3) and AFM (Fig. 4). The discussion states that "AFM is less sensitive to numbers for large aggregates (100 nm and above)". What is the reason? I think it's better to state the reason.

Minor comments

1. Figure 1B. Please add a scale bar to the electron microscope image of 10-40% sucrose.
2. Figure 2. It is better to add the statistical description of the main text lines 169 to 171 to the legend.
3. Does the histogram in Figure 2 cover every pixel in each image?
4. At first glance, it is difficult to tell whether the histogram in Figure 2 corresponds to the author's claim although if he/she focus on the area larger than 1 on the horizontal axis, it can see that it matches the

author's claim. It would be clearer to make a line graph and combine the data for all sucrose concentrations into one graph.

5. Figure 10 is missing

Reviewer #2 (Remarks to the Author):

Please see attached PDF.

Reviewer #3 (Remarks to the Author):

In this article, Derya Emin and colleagues propose that small soluble aggregates of α -synuclein constitute the toxic forms in PD. They apply a non-perturbing method of discontinuous sucrose gradient to separate different species of α -synuclein formed in an in vitro aggregation reaction and characterize the structural properties using TIRFM, single-molecule super-resolution microscopy, AFM, and SimPull. Small α -synuclein aggregates are shown to induce more TNF- α release from BV2 cells and higher membrane permeability in POPC liposomes. Importantly, the authors find that soluble aggregates extracted from post-mortem PD brains exhibit similar structural characteristics as the small aggregates formed in the in vitro reaction and are smaller and more inflammatory than controls. Therefore, they suggest that small non-fibrillar α -synuclein species critically contribute to the pathogenesis of PD.

The identification of toxic species of α -synuclein is of great interest to the field of PD research. These species may serve as potential drug targets of disease-modifying therapies. Most of the experiments in this study appear to have been conducted carefully with appropriate controls.

My major concern relates to the toxicity assays performed here. Effects of these aggregates on neurons are not accessed, which is crucial for our understanding of the pathogenesis in PD. Furthermore, two variables (species and length) are involved in these sucrose fractions, both of which are known to affect the cytotoxicity. It was reported that short α -synuclein fibrils of 50 nm or less are most efficient in promoting phosphorylated α -synuclein in cultured cells and mouse models (doi: 10.1074/jbc.M116.734707). Therefore, I recommend the authors to compare the toxicity with species of the same length and with the same species of different lengths. Previous in vivo studies showed that α -synuclein fibrils lead to a sustained progressive loss of dopamine neurons compared to oligomeric

species in rats and mice (doi: 10.1007/s00401-011-0926-8; doi: 10.1074/jbc.RA119.007743). The authors should at least discuss these results.

I am also concerned that the postmortem intervals of the brains are highly variable with one PD brain being 108 hours. Three PD brains and three controls are included in this study. Shorter, similar postmortem intervals and more samples would be more convincing.

The authors only compare the soluble aggregates from PD brains with controls. However, it remains unknown whether the insoluble aggregates exert higher toxicity and contribute more to the disease onset and progression. It is necessary to study both fractions quantitatively. In addition, immunodepletion of A β and α -synuclein should be done to distinguish these aggregates and their toxicity.

REVIEWER COMMENTS

Reviewer #1 (Remarks to the Author):

Drs. Emin and Klenerman's group showed that aggregates of wild-type α -synuclein (α syn) smaller than 200 nm in length, cause inflammation and permeabilization of liposome membranes. Studying soluble aggregates extracted from post-mortem human brains also reveals that these aggregates are similar in size and structure to the smaller aggregates. Furthermore, the soluble aggregates present in PD brains are smaller, largely less than 50 nm, and more inflammatory compared to the larger aggregates present in control brains. This study suggests that the small non-fibrillar α syn aggregates are the critical species in the pathogenesis of PD.

Major comments

1. How similar or different are the symptoms of the three PD patients? It would be nice to have an explanation as to whether the toxic aggregate sizes are similar for different symptoms.

The patients have been diagnosed with Parkinson's disease based on their neuropathology and the presence of Lewy bodies. They all had advanced Parkinson's disease (disease duration 12-19 years). We do not have further information regarding their clinical state prior to death.

2. Can the difference in length and shape between EM, AFM and observation using the patient's brain be explained only by the difference in the methods? What causes the difference of 50 nm and 200 nm in *in vitro* and actual brain systems.

We have added a more detailed discussion of the reason for the differences between AFM and EM. This is essentially due to the mica surface capturing the smaller aggregates more effectively than larger fibrils. The reason for the difference in the aggregates formed in *in vitro* and in brains is probably due to the conditions of the aggregation reaction. Please note that we aggregate α -synuclein at 70 μ M using a standard protocol leading to aggregates in a time scale of hours. In contrast, in the brain this is a process which takes several decades in the presence of cellular mechanisms to slow-down and remove aggregates.

3. It is better to describe how long it takes to measure the aggregate size after separation by sucrose density gradient centrifugation. If possible, it is better to describe the presence or absence of changes in size after separation.

The aggregates are separated and then frozen in aliquots and then thawed and imaged at a later stage. We have clarified this point in the revised paper.

We have not studied the changes in aggregate size after separation because we immediately snap freeze the sucrose fractions after separation and then thaw an individual aliquot for selected measurements. However, we have done experiments on synthetic and brain-derived aggregates to check their stability. These are outlined below.

In our previous work synthetic aggregates of abeta and α -synuclein were characterised immediately after formation and then over 24 hours to check that the aggregates are stable and that the monomer that is also present in the preparation does not form additional aggregates. ¹ We found that for α -synuclein that there was no change in the number of aggregates over 24 hours, due to continued aggregation, provided the monomer concentration was below 1 μ M. ² For abeta₄₂ there was not change in the number of aggregates over 24 hour provided the monomer was less than 10 nM. ³

In our early work on synthetic α -synuclein aggregates, we showed that there was no decrease in aggregate concentration on dilution by a factor of 100,000 and no change in structure based on FRET. ⁴ These signals also do not change over 3 hours during single-molecule imaging. We have also checked that there is no reduction in aggregate number or structure when aliquoting and then freezing and thawing samples of synthetic aggregates, using single-molecule fluorescence. Centrifugation using sucrose gradients has been used by us and others to separate brain-derived ⁵ and synthetic aggregates of tau ⁶ and synthetic aggregates of abeta ⁷ based on their size showing that the aggregates are stable to centrifugation.

There is also a large body of evidence that brain-derived aggregates are stable once formed and maintain their toxicity, like synthetic aggregates discussed above. A recent study on oligomeric α -synuclein in CSF has also shown that oligomeric α -synuclein levels remained relatively stable over multiple tube transfers and upon delayed storage, but decreased when subject to multiple freeze-thaw cycles. ⁸ Selkoe and co-workers showed that abeta aggregates recovered from homogenised human tissue samples produced LTP deficit in brain slices. ⁹ More recently, Walsh and co-workers showed that toxic aggregates could be recovered from soaked brain tissue, without the need for any homogenisation and maintained their toxicity causing LTP deficit ¹⁰ and neurite retraction ¹¹ after freezing for storage and then thawing for experiments.

4. Does the aggregate that has been structurally observed after separation cause cytotoxicity? The authors should examine whether all aggregates are still small in size after incubation time in the cell culture system.

We used 500 nM total monomer in these experiments and exchanged the buffer every 24 hours. We previously checked if oligomers are formed during the incubation with cells for 24 h before buffer is exchanged and found that for incubations below 1000 nM monomer and 15 nM oligomers, there is no significant change in oligomer concentration over the 24-h incubation. ¹ At these low concentrations of monomer there will also be no significant monomer addition. Therefore, this data shows that there is no significant change in the aggregates over the 24 hours incubation under these conditions.

We have cited this paper and made this point in the revised manuscript. We have also done an additional experiment to confirm this using sonicated fibrils which range in size from 30 nm to 600 nm. We imaged these aggregates using super-resolution imaging at the start of the experiment and then after 24 hours exposure to macrophage, see figure below. There was no significant difference in the size distribution of the aggregates as shown in the cumulative frequency plot of the aggregates at 0 hours and after 24 hours exposed to macrophage. There was also no significant difference between the fraction of aggregates less than 100 nm in size.

5. The shape (linearity) of the aggregates contained in the fraction with high sucrose concentration appears to be different between the super-resolution fluorescence microscope (Fig. 3) and AFM (Fig. 4). The discussion states that "AFM is less sensitive to numbers for large aggregates (100 nm and above)". What is the reason? I think it's better to state the reason.

There are two main reasons. The first is related to the different spatial resolution and field of view of AFM and super-resolution fluorescence. AFM images much smaller regions and at much higher resolution than super-resolution fluorescence.⁷ In particular, AFM allows to measure down to a single a-synuclein monomer (0.3 nm height, ca. 10 nm convoluted diameter), while super-resolution fluorescence has a larger field of view but with smaller spatial resolution of ca. 30 nm. The second

reason is the different surface of deposition leading to partial differential absorption. In this study, samples for AFM analysis were deposited on a negative mica surface which is ideal to study soluble aggregates and oligomers, although some larger aggregates are visible (SI Fig. 2), the surface have less affinity to larger aggregates.¹²⁻¹⁴

So, overall, in AFM is more likely to image and measure small aggregates rather than large aggregates in the sample, which causes a shift of the average of the size distributions towards significantly lower values than super-resolution fluorescence. We have explained this in the revised discussion.

Minor comments

1. Figure 1B. Please add a scale bar to the electron microscope image of 10-40% sucrose.

This has been done.

2. Figure 2. It is better to add the statistical description of the main text lines 169 to 171 to the legend.

This has been done.

3. Does the histogram in Figure 2 cover every pixel in each image?

The histogram is for every aggregate detected. We imaged at least 83 fields of view and combined all the data (N= 105 for 20%, N= 96 for 30%, N= 83 for 40% and N= 94 for 50%). The figure just shows representative images. We have added how many aggregates were used to make up the histograms in the figure legend

4. At first glance, it is difficult to tell whether the histogram in Figure 2 corresponds to the author's claim although if he/she focus on the area larger than 1 on the horizontal axis, it can see that it matches the author's claim. It would be clearer to make a line graph and combine the data for all sucrose concentrations into one graph.

We are not sure what the reviewer is requesting here. We have added a zoom to make the differences clearer.

5. Figure 10 is missing

This was a mistake and there is no Figure 10.

Reviewer #2 (Remarks to the Author):

Review of Nature Communications manuscript NCOMMS-22-52057-T

“Small soluble α -synuclein aggregates are the toxic species in Parkinson’s disease”

In this paper, Emin et al. study the toxicity of α -synuclein correlated with aggregate size, structure, and morphology. The authors separate the α -synuclein aggregates using sucrose density centrifugation. Various complementary techniques are used to measure the size and structure of the aggregates, including transmission electron microscopy (TEM), total internal reflection fluorescence (TIRF) microscopy of thioflavin T (ThT), single-molecule localization microscopy (SMLM) using thioflavin X (ThX), atomic force microscopy (AFM), single-molecule pulldown (SimPull) and imaging, and aptamer DNA PAINT (AD-PAINT). The authors also use MJF and SC antibodies to show that aggregates isolated from 10% sucrose lack β -sheet structures, which is in agreement with super-resolution (SR) measurements. Analyzing the inflammatory response of BV2 mouse microglia to different sizes of aggregates, the authors conclude that the smaller aggregates are more toxic compared to fibrils. They also show that the aggregates extracted from PD patients are smaller and produce greater inflammatory response compared to those isolated from human control brains.

Overall, I believe the paper is easy to read and provides valuable insights into the toxicity of α -synuclein that small non-fibrillar α -synuclein aggregates are critical species driving neuroinflammation and disease progression. The authors present a compelling series of experiments on samples collected from human patients that support their conclusions. This work should be of interest to the broad readership of Nature Communications. However, there are some issues related to experimental details and presentation that need to be addressed before publication.

Detailed recommendations for improvement are below.

1. Could the authors comment on the 40% and 50% panels in Figure 1B, specifically, the morphology and width of the fibril-like structures? Are these truly representative of the aggregates found at the labeled sucrose concentrations? The lengths and widths of the structures shown here seem to be very different from those measured by SMLM and AFM.

The TEM images resemble the SMLM images but have higher resolution. The resolution of the SMLM images is about 30 nm so the fibrils appear thicker. As discussed above the negative mica surface used for AFM captures smaller aggregates better than larger aggregates. However, the morphology of the fibrils measured by TEM in fig. 2 is similar to the ones measured by AFM in Fig. S1.

2. The data presented in Figure 2 are difficult to interpret:

(a) The histograms of intensities for different sucrose levels are difficult to distinguish from one another. According to Table 1, the difference in median is within 0.1, which is not readable on the log-scale intensity axis. I recommend plotting instead the cumulative distributions of intensity similar to that in Figure 3B so the changes are easier to see.

We do plot the cumulative distribution of intensities but have now included a zoom to make the differences clearer.

(b) From the images shown, it seems like there should be two clear populations of pixel intensities: one associated with the aggregates themselves and another associated with the dark background. Why can't we observe the dark background level in the histograms? Is it because of the log-scale cutoff of 0.01?

This is because we only use data above a certain threshold determined by the background level and background noise.

(c) It is difficult to connect the histograms to the images without a quantitative color scale.

Please include color scales so that the images can be interpreted quantitatively.

A colour scale has been added.

3. Details within some of the images in the paper are difficult to see, e.g., the TIRF images in Figure 2 and the AFM images in Figure 4. Could the authors include some representative zooms of 1 individual aggregates similar to those in Figure 3?

Zooms have now been added.

4. For the zoomed examples shown in Figure 3, please label their measured heights and diameters.

We have added the length and diameter to all the zoomed images in Figure 3.

5. The authors should comment on the lengths of the representative aggregates shown in Figure 3D-G. Most of them, especially for 40% and 50% sucrose levels, are longer than the median lengths shown in Figure 3A. Can the authors show truly representative fibrils instead of outliers? Please also give the measured lengths of each example aggregate shown in the figure.

Similarly the representatives in Figure 8J-L do not match the length distributions shown in Figure 8E – all are above average. Please show representative aggregates.

We have added more representative images.

6. The authors should include how they evaluate and interpret eccentricity in the paper; there are not enough details.

(a) Please clarify exactly how the eccentricity of an aggregate is calculated from the SMLM data.

(b) Many of the aggregates in Figure 3D (20% sucrose) seem to be almost spherical, yet their mean eccentricity is 0.66. Please label the measured eccentricities for the aggregates shown in Figure 3D-G.

We have added the details of the analysis of the single molecule super-resolution data to the methods and explain how we determined the eccentricity. The eccentricity of a cluster was determined by fitting an ellipse to the cluster and determining the focal distance of the ellipse divided by the maximum distance of the major axis. We also checked our code following the reviewer's comment and as a result of finding an error have carefully reanalysed all the data and as a result the eccentricity values and length values have changed. Thank you for spotting this mistake.

7. The authors seem to use “cross-sectional diameter” and “convoluted diameter” interchangeably, e.g., on line 198 and in Figure 4. Please clarify how these terms should be interpreted and how they are calculated from the AFM data.

Yes, the terms “cross-sectional diameter” and “convoluted diameter” are equivalent. The diameter of each biomolecules is directly measured from the raw data without any further processing. AFM acquires 3-D maps of morphology of biomolecules on a surface. By taking the cross-section along the Z direction with a line profiles of each biomolecule on the surface it is thus possible to measure their cross-sectional height and diameter.¹⁵ The measurement of height is extremely precise at the angstroms level.⁷ Instead, the measurement of the diameter of biomolecules with similar size to the AFM tip (<8-10 nm) are always subjected to an overestimation of the value, which is well known and called “convolution” effect¹⁶, we specified this effect to help audience from other fields to interpret the data.

8. Lines 293-299 contain references to “Figure 10A” and “Figure 10A-C,” but this figure does not exist. Some of the data seem to be contained in Figure 8, but I’m not sure. Please fix these references.

These references are fixed.

9. The Kolmogorov-Smirnov test is utilized several times in this paper, but more details should be provided:

(a) Please state the number of aggregates used for the test in Figure 8C.

(b) Similarly, please quantify the number of aggregates used for the test in Figure 9A.

(c) Please also quantify the number of aggregates used for the test in Supplementary Figure 8A.

We have added these details.

10. Since SimPull-STORM was used to quantify the data in Supplementary Figure 8, it would be nice to see some representative examples of fibers from the PD and HC populations.

We have added these images.

11. Can the authors comment on the use of thioflavin T and thioflavin X vs. other amyloidophilic dyes, such as Congo red, Nile red, etc.? For example, Ref. 42 showed that Nile red was more sensitive for detecting small oligomers and proto-fibrils. It seems to me that the SMLM analysis would be even more powerful if other dyes were used.

This is a good point and certainly it is possible to use additional dyes. The main idea of the study was to use methods that could be applied to both the synthetic aggregates and the brain derived aggregates, so we would need to explore how well these other dyes work on the brain-derived samples where other material will also be present that these dyes can bind to.

12. Line 547: Please clarify the meaning of F in the formula.

This has been done.

13. Lines 563-564: Please clarify how the DNA-conjugated aptamers were synthesized/obtained.

This has been added.

14. Table 4: It would be helpful to change "647-labelled" to "Alexa647-labelled" in the table.

This has been done.

15. Several citations are missing that would be helpful to give more background and context for this work:

(a) “Transient amyloid binding” using thioflavin T was first shown by Spehar, et al., and this work should be cited on line 175: Spehar, K., Ding, T., Sun, Y., Kedia, N., Lu, J., Nahass, G. R., Lew, M. D., and Bieschke, J. Super-Resolution Imaging of Amyloid Structures over Extended Times by Using Transient Binding of Single Thioflavin T Molecules. *ChemBioChem*. 19, 1944–1948 (2018). doi:10.1002/cbic.201800352

We have added this reference.

(b) The acronym and citation of the initial demonstration of AD-PAINT were not included when the technique is first mentioned in line 260. These details should be included.

Whiten, D. R., Zuo, Y., Calo, L., Choi, M.-L., De, S., Flagmeier, P., Wirthensohn, D. C., Kundel, F., Ranasinghe, R. T., Sanchez, S. E., Athauda, D., Lee, S. F., Dobson, C. M., Gandhi, S., Spillantini, M.-G., Klenerman, D., and Horrocks, M. H. Nanoscopic Characterisation of Individual Endogenous Protein Aggregates in Human Neuronal Cells. *ChemBioChem*. 19, 2033–2038 (2018). doi:10.1002/cbic.201800209

This has been added.

16. There are several typographical errors that should be corrected:

(a) Line 332: “size, morphology” should be “size and morphology”.

(b) Line 544: The “2” should be in superscript in “10 W/cm²”.

(c) Line 599: “EMCDD” should be “EMCCD”.

(d) Lines 609-610: “...for 1h;” is missing a semicolon. Similarly, “...was repeated.” is missing a period at the end of the sentence.

(e) Line 616: A stylistic comment. Beginning the sentence with the numeral “3” makes it more difficult to read. “Three” would be better.

(f) Line 985: Should “cell” be “liposome” here?

(g) Line 986: “different!” should be “different”.

These have all been corrected. Thank you.

Reviewer #3 (Remarks to the Author):

In this article, Derya Emin and colleagues propose that small soluble aggregates of α -synuclein constitute the toxic forms in PD. They apply a non-perturbing method of discontinuous sucrose gradient to separate different species of α -synuclein formed in an in vitro aggregation reaction and characterize the structural properties using TIRFM, single-molecule super-resolution microscopy, AFM, and SimPull. Small α -synuclein aggregates are shown to induce more TNF- α release from BV2 cells and higher membrane permeability in POPC liposomes. Importantly, the authors find that soluble aggregates extracted from post-mortem PD brains exhibit similar structural characteristics as the small aggregates formed in the in vitro reaction and are smaller and more inflammatory than controls. Therefore, they suggest that small non-fibrillar α -synuclein species critically contribute to the pathogenesis of PD.

The identification of toxic species of α -synuclein is of great interest to the field of PD research. These species may serve as potential drug targets of disease-modifying therapies. Most of the experiments in this study appear to have been conducted carefully with appropriate controls.

My major concern relates to the toxicity assays performed here. Effects of these aggregates on neurons are not accessed, which is crucial for our understanding of the pathogenesis in PD.

The concentration of aggregates is very low in the soaked brain samples making performing toxicity assays very challenging. In this work we used two assays that we had already developed and had sufficient sensitivity to show detectable changes. These are an assay to sensitively measure the presence of aggregates that can cause membrane permeabilisation, and hence lead to disrupted calcium ion homeostasis, using liposomes, and an inflammatory assay using a microglial cell-line. We do not have a working highly sensitive assay for toxicity to neurons and this would take considerable development. There are two other possible assays used by Walsh and colleagues. Abeta aggregates cause long-term potentiation deficit in brain slices. This is effectively a digital assay with the sample either causing a deficit or not and requires a large amount of sample. This is a cellular correlate of memory loss and hence relevant to Alzheimer's disease but less obviously relevant to PD. The other assay is a neurite retraction assay which has been used for abeta aggregates but has not been used for α -synuclein aggregates, so this would need to be developed and tested that α -synuclein aggregates cause retraction before being applied to the soaked brain samples.

We agree with the reviewer for the need for more sensitive assays and assays that work on neurons, in addition to glial cell assays. To address this point we have added a paragraph discussing these limitations and the need to develop more sensitive assays and assays that work on neurons to the revised manuscript.

Furthermore, two variables (species and length) are involved in these sucrose fractions, both of which are known to affect the cytotoxicity. It was reported that short α -synuclein fibrils of 50 nm or less are most efficient in promoting phosphorylated α -synuclein in cultured cells and mouse models (doi: 10.1074/jbc.M116.734707). Therefore, I recommend the authors to compare the toxicity with species of the same length and with the same species of different lengths.

It is not possible to obtain homogeneous samples of aggregates of a particular size and structure. Even in the paper cited the fibrils used for seeding were a range of sizes. At present the separation that is performed is the best possible and provided fractions of a certain size range. There will be some correlation between the size and structure of the aggregates since larger aggregates will be fibrils but the separation is only based on size and not species/structure.

Previous in vivo studies showed that α -synuclein fibrils lead to a sustained progressive loss of dopamine neurons compared to oligomeric species in rats and mice (doi: 10.1007/s00401-011-0926-8; doi: 10.1074/jbc.RA119.007743). The authors should at least discuss these results.

Thank you for this reference which we have added. The data in this paper suggests that adding fibrils that can replicate is very damaging but oligomers are also toxic. We have also added the point that it is hard to extrapolate from experiments performed where high concentrations of a particular type of aggregate are formed to what happens in vivo where the number of aggregates and the distribution of their sizes and structures will be different in the introduction. This is the advantage of using the soaked brain samples but it raises challenges about having sufficiently sensitive toxicity assays. We have also added this to the discussion.

I am also concerned that the postmortem intervals of the brains are highly variable with one PD brain being 108 hours. Three PD brains and three controls are included in this study. Shorter, similar postmortem intervals and more samples would be more convincing.

This current work on the post mortem tissue is essentially a pilot study aiming to show that it is possible to characterize the aggregates from human brains and make some measurements of their toxicity. Whilst a larger sample size and shorter post mortem intervals would be preferable, post mortem tissue availability is limited and the interval between death and tissue collection is typically highly variable. Our post mortem intervals are comparable to those in large neuropathological studies.¹⁷ Importantly, mean post mortem interval was similar in the PD group versus the control group (mean for controls, 42.7 hours, mean for PD, 55 hours) and we found no significant correlation between aggregate length and PM interval (A, Pearson R= 0.29) nor age (B, Pearson R=0.06) across the six brains. Nonetheless, we acknowledge that it would be preferable to minimise the variability in PM intervals and increase the number of brain samples in future larger studies. We have added this point to the discussion.

A**Correlation Age and Aggregate Length****B****Correlation between Post-Mortem Interval and Length**
The authors only compare the soluble aggregates from PD brains with controls. However, it remains unknown whether the insoluble aggregates exert higher toxicity and contribute more to the disease onset and progression. It is necessary to study both fractions quantitatively.

In addition, immunodepletion of A β and α -synuclein should be done to distinguish these aggregates and their toxicity.

We used a very gentle method to extract the soluble toxic aggregates based on soaking the brain tissue based on published work by Walsh and colleagues. The aggregates extracted by soaking were about 25 % of all aggregates but contained all the species found to be toxic using LTP deficit and neurite retraction. Furthermore, the problem is that to extract the insoluble aggregates a homogenization step is needed where the larger aggregates are broken up into smaller aggregates potentially altering their properties.

We agree with the reviewer that we are limited in our toxicity assays but this is because the brain-derived aggregates are present at low concentrations and more sensitive assays are needed. Our approach in the paper was to establish the size range of toxic synthetic aggregates. We can then determine if similar size aggregates are present in brain and if they differ between HC and PD patients as well as measure the toxicity with the best available assays. The neuroinflammation assay showed that there was an inflammatory response but it does not have a sufficient dynamic range to detect smaller changes when we immunodeplete the A β and α -synuclein aggregates. A more sensitive neuroinflammatory assay is needed.

These are all good suggestions from the reviewer but unfortunately these experiments are not feasible at present. We therefore discuss these limitations in a new paragraph in the discussion.

References

1. C. D. Hughes *et al.*, Picomolar concentrations of oligomeric alpha-synuclein sensitizes TLR4 to play an initiating role in Parkinson's disease pathogenesis. *Acta Neuropathologica* **137**, 103-120 (2019).
2. C. D. Hughes *et al.*, Picomolar concentrations of oligomeric alpha-synuclein sensitizes TLR4 to play an initiating role in Parkinson's disease pathogenesis. *Acta Neuropathol* **137**, 103-120 (2019).
3. C. Hughes *et al.*, Beta amyloid aggregates induce sensitised TLR4 signalling causing long-term potentiation deficit and rat neuronal cell death. *Commun Biol* **3**, 79 (2020).
4. N. Cremades *et al.*, Direct observation of the interconversion of normal and toxic forms of α -synuclein. *Cell* **149**, 1048-1059 (2012).
5. S. Maeda *et al.*, Increased levels of granular tau oligomers: An early sign of brain aging and Alzheimer's disease. *Neuroscience Research* **54**, 197-201 (2006).
6. S. Maeda *et al.*, Granular tau oligomers as intermediates of tau filaments. *Biochemistry* **46**, 3856-3861 (2007).
7. S. De *et al.*, Different soluble aggregates of A β 42 can give rise to cellular toxicity through different mechanisms. *Nature Communications* **10**, 1541 (2019).
8. I. Y. Abdi *et al.*, Preanalytical Stability of CSF Total and Oligomeric Alpha-Synuclein. *Frontiers in Aging Neuroscience* **13**, (2021).
9. D. M. Walsh *et al.*, Naturally secreted oligomers of amyloid beta protein potently inhibit hippocampal long-term potentiation in vivo. *Nature* **416**, 535-539 (2002).
10. W. Hong *et al.*, Diffusible, highly bioactive oligomers represent a critical minority of soluble A β in Alzheimer's disease brain. *Acta neuropathologica* **136**, 19-40 (2018).
11. M. Jin *et al.*, An in vitro paradigm to assess potential anti-A β antibodies for Alzheimer's disease. *Nature Communications* **9**, 2676 (2018).
12. S. Ruggeri Francesco *et al.*, Identification and nanomechanical characterization of the fundamental single-strand protofilaments of amyloid α -synuclein fibrils. *Proceedings of the National Academy of Sciences* **115**, 7230-7235 (2018).
13. F. S. Ruggeri *et al.*, Influence of the β -Sheet Content on the Mechanical Properties of Aggregates during Amyloid Fibrillization. *Angewandte Chemie International Edition* **54**, 2462-2466 (2015).
14. F. S. Ruggeri *et al.*, The Influence of Pathogenic Mutations in α -Synuclein on Biophysical and Structural Characteristics of Amyloid Fibrils. *ACS Nano* **14**, 5213-5222 (2020).
15. F. S. Ruggeri, T. Šneideris, M. Vendruscolo, T. P. J. Knowles, Atomic force microscopy for single molecule characterisation of protein aggregation. *Archives of Biochemistry and Biophysics* **664**, 134-148 (2019).
16. F. Marques-Moros, A. Forment-Aliaga, E. Pinilla-Cienfuegos, J. Canet-Ferrer, Mirror effect in atomic force microscopy profiles enables tip reconstruction. *Scientific Reports* **10**, 18911 (2020).

17. Kouli, A., Camacho, M., Allinson, K. & Williams-Gray, C. H. Neuroinflammation and protein pathology in Parkinson's disease dementia. *Acta Neuropathol. Commun.* **8**, 1–19 (2020).

REVIEWERS' COMMENTS

Reviewer #1 (Remarks to the Author):

The revised MS has been much improved. However, the reviewer has still a few concerns.

As the authors also know, the heterogeneity of aggregate structure, pathology, and symptoms of actual patients is very important for thinking about the significance of this paper. Please consider examining whether these detailed differences might reflect size and toxicity in the experimental system.

And the authors should explain why unstable oligomers are able to be stable during the incubation time for 24 hrs; the concentration is lower than the critical concentration.

Reviewer #3 (Remarks to the Author):

The authors have addressed all of my concerns in the revision. I have no further comments.

Response to reviewer

Reviewer #1 (Remarks to the Author):

The revised MS has been much improved. However, the reviewer has still a few concerns. As the authors also know, the heterogeneity of aggregate structure, pathology, and symptoms of actual patients is very important for thinking about the significance of this paper. Please consider examining whether these detailed differences might reflect size and toxicity in the experimental system.

As we stated in our previous response, we do not have additional data on the patient symptoms so cannot examine if there are differences that can be explained by the aggregate size or toxicity. However, we agree this is a good point so have added a sentence suggesting this is also done in future work.

And the authors should explain why unstable oligomers are able to be stable during the incubation time for 24 hrs; the concentration is lower than the critical concentration.

There is a body of published work that shows the α -synuclein aggregates are stable once formed, and hence there is negligible dissociation over 24 hours, even if the monomer concentration has been reduced by dilution below the critical concentration. For example we found no dissociation of aggregates over several hours when diluted down to picomolar concentrations for single molecule analysis (Cell 149, 1048–1059 (2012)) We have now added a sentence stating this and referencing the relevant published work.